

# A new discrete wavelength BUV algorithm for consistent volcanic SO₂ retrievals from multiple satellite missions

Bradford L. Fisher[1], Nickolay A. Krotkov[2], Pawan K. Bhartia[2], Can Li[2,3], Simon Carn[4], Eric Hughes[5] and Peter J. T. Leonard[6]

[1]SSAI, 10210 Greenbelt Rd, Suite 600, Lanham MD, 20706, USA

[2]Atmospheric Chemistry and Dynamics Laboratory, NASA Goddard Space Flight Center, Greenbelt MD, 20771, USA

[3] Earth System Science Interdisciplinary Center, University of Maryland, College Park, MD, 20742, USA

[4] Geological and Mining Engineering and Sciences, Michigan Technological University, Houghton MI 49931, USA

[5] Miner & Kasch, 8174 Lark Brown Rd, #101, Eldridge, MD 21075.

[6] ADNET Systems, Inc., 7515 Mission Drive, Suite A100, Lanham, MD 20706

*Correspondence to*: Bradford.Fisher (bradford.fisher@ssaihq.com)

**Abstract.** This paper describes a new discrete wavelength algorithm developed for retrieving volcanic sulfur dioxide (SO₂) vertical column density (VCD) from UV observing satellites. The Multi-Satellite SO₂ algorithm (MS_SO2) simultaneously retrieves column densities of sulfur dioxide, ozone,

Lambertian Effective Reflectivity (LER) and its spectral dependence. It is used operationally to process measurements from the heritage Total Ozone Mapping Spectrometer (TOMS) on board NASA's Nimbus-7 satellite (N7/TOMS: 1978-1993) and from the current Earth Polychromatic Imaging Camera (EPIC) on board Deep Space Climate Observatory (DSCOVR: 2015-) from the Earth-Sun Lagrange (L1) orbit. Results from MS_SO2 algorithm for several volcanic cases were validated using the more

sensitive principal component analysis (PCA) algorithm. The PCA is an operational algorithm used by NASA to retrieve SO₂ from hyperspectral UV spectrometers, such as Ozone Monitoring Instrument (OMI) on board NASA's Earth Observing System Aura satellite and Ozone Mapping and Profiling Suite (OMPS) on board NASA-NOAA Suomi National Polar Partnership (S-NPP) satellite. For this comparative study, the PCA algorithm was modified to use the discrete wavelengths of the

Nimbus7/TOMS instrument. Our results demonstrate good agreement between the two retrievals for the largest volcanic eruptions of the satellite era, such as 1991 Pinatubo eruption. To estimate SO₂ retrieval



uncertainties we use radiative transfer simulations explicitly accounting for volcanic sulfate and ash aerosols. Our results suggest that the discrete-wavelength MS_SO2 algorithm, although less sensitive than hyperspectral PCA algorithm, can be adapted to retrieve volcanic SO2 VCDs from contemporary hyperspectral UV instruments, such as OMI and OMPS, to create consistent, multi-satellite, long-term
volcanic SO2 climate data records.

# 1 Introduction

Volcanic eruptions are an important natural driver of global climate change, but unlike other natural climate forcing (e.g., changes in Earth's orbit, solar irradiance), the magnitude of volcanic forcing is highly variable, largely unpredictable, and the effects are typically more transient. Of most interest are
the episodic, large injections of volcanic sulfur dioxide ($SO_2$) into the Earth's stratosphere by major explosive volcanic eruptions, the most recent example being the eruption of Pinatubo (Philippines) in June 1991 (e.g., Bluth et al., 1992; Guo et al., 2004). Stratospheric loading of volcanic $SO_2$ by major eruptions leads to the formation of sulfuric acid (or sulfate) aerosols that scatter incoming solar shortwave radiation and absorb outgoing thermal radiation over timescales of months to years, cooling
the troposphere and warming the stratosphere (e.g., Robock, 2000). Primary volcanic emissions of aerosols such as volcanic ash can also have atmospheric and climate impacts, but these are typically more short-lived. Volcanic eruptions can also release reactive halogen species into the atmosphere, such as chloride and bromide (Mankin and Coffey, 1984; Bobrowski et al., 2003; Kern et al., 2008). Halogens can impact the total column ozone amount and profile shape if injected into the lower
stratosphere (Solomon et al., 1998, Klobas et al., 2017), but sulfate aerosols are also required to catalyze the heterogeneous chemical reactions that can efficiently deplete ozone. Hence, to understand the impacts of volcanic eruptions on climate, and in order to predict possible outcomes in the event of a major eruption, long-term satellite measurements of volcanic $SO_2$ emissions are essential.

The satellite record of volcanic $SO_2$ emissions by major volcanic eruptions extends back to
1978, and has been derived from instruments operating in both the ultraviolet (UV) and infrared (IR) spectral regions (Fig. 1; e.g., Carn et al., 2003, 2016, 2019; Prata et al., 2003). Measurements in the UV have a longer heritage, since the first satellite detection of volcanic $SO_2$ was achieved by the UV Total Ozone Mapping Spectrometer (TOMS) in 1982 following the eruption of El Chichon (Mexico; Krueger, 1983; Krueger et al., 2008), and interference from volcanic $SO_2$ must be accounted for in order to
produce accurate, long-term UV measurements of ozone. UV measurements have greater sensitivity to the total atmospheric $SO_2$ column than IR retrievals and hence the former have been the mainstay of volcanic $SO_2$ monitoring during the satellite era to date. The volcanic $SO_2$ climatology from 1978-present (Fig. 1, Carn 2019) reveals highly variable inter-annual volcanic $SO_2$ forcing dominated by two



major eruptions (El Chichon in 1982 and Pinatubo in 1991), with the post-2000 period dominated by smaller eruptions. Although none of these smaller eruptions have, individually, produced measurable climate effects, collectively they have garnered significant interest as they may play an important role in sustaining the persistent, background stratospheric aerosol layer, which is an important factor in global
climate forcing (e.g., Solomon et al., 2011; Vernier et al., 2011; Ridley et al., 2014).

One of the key challenges in assembling a long-term, consistent, satellite-based volcanic $SO_2$ emissions climatology (e.g., Fig. 1) is merging measurements from sensors with different spectral coverage and resolution. This complicates any analysis of 'trends' in volcanic $SO_2$ loading (e.g., in the post-2000 period of moderate volcanic eruptions; Fig. 1) or comparisons of eruptions of similar
magnitude in different satellite instrumental eras. A step change in $SO_2$ sensitivity occurred when the multi-spectral, six-channel TOMS instruments were superseded by hyperspectral UV sensors, such as the Global Ozone Monitoring Experiment (GOME, 1995-2003; Khokhar et al., 2005), the Scanning Imaging Absorption Spectrometer for Atmospheric Chartography (SCIAMACHY, 2002-2012; Lee et al., 2008), the Ozone Monitoring Instrument (OMI, 2004- ; Krotkov et al., 2006) and the Ozone
Mapping and Profiler Suite (OMPS, 2012- ; Carn et al., 2015). This is manifested in Figure 1 by an increased number of detected volcanic eruptions with low $SO_2$ loading (<10 kt) after 2004 (note that GOME and SCIAMACHY measurements are not shown in Fig. 1), whereas rates of global volcanic activity have not changed significantly. UV $SO_2$ retrieval algorithms have also evolved substantially since the 1980s in response to advances from multi-spectral to hyperspectral sensors, improvements in
ozone retrievals, and efforts to account for volcanic ash and aerosol interference (e.g., Krueger et al., 1995, 2000; Krotkov et al., 1997, 2006; Yang et al., 2007, 2010; Li et al., 2013, 2017; Theys et al., 2015). However, to date there has been no attempt to develop a single algorithm that could be used to generate a long-term, consistent $SO_2$ climatology across multiple UV satellite missions. In this paper we describe a new Multi-Satellite $SO_2$ algorithm (MS_SO2) that is applicable to both multi-spectral (e.g.,
TOMS) and hyperspectral (e.g., OMI) UV measurements. As a first step in the generation of a multi-satellite volcanic $SO_2$ record, we apply the MS_SO2 algorithm to the Nimbus-7 TOMS (N7/TOMS) measurements (1978-1993) and present a reanalysis of some of the most significant eruptions of the N7/TOMS mission.





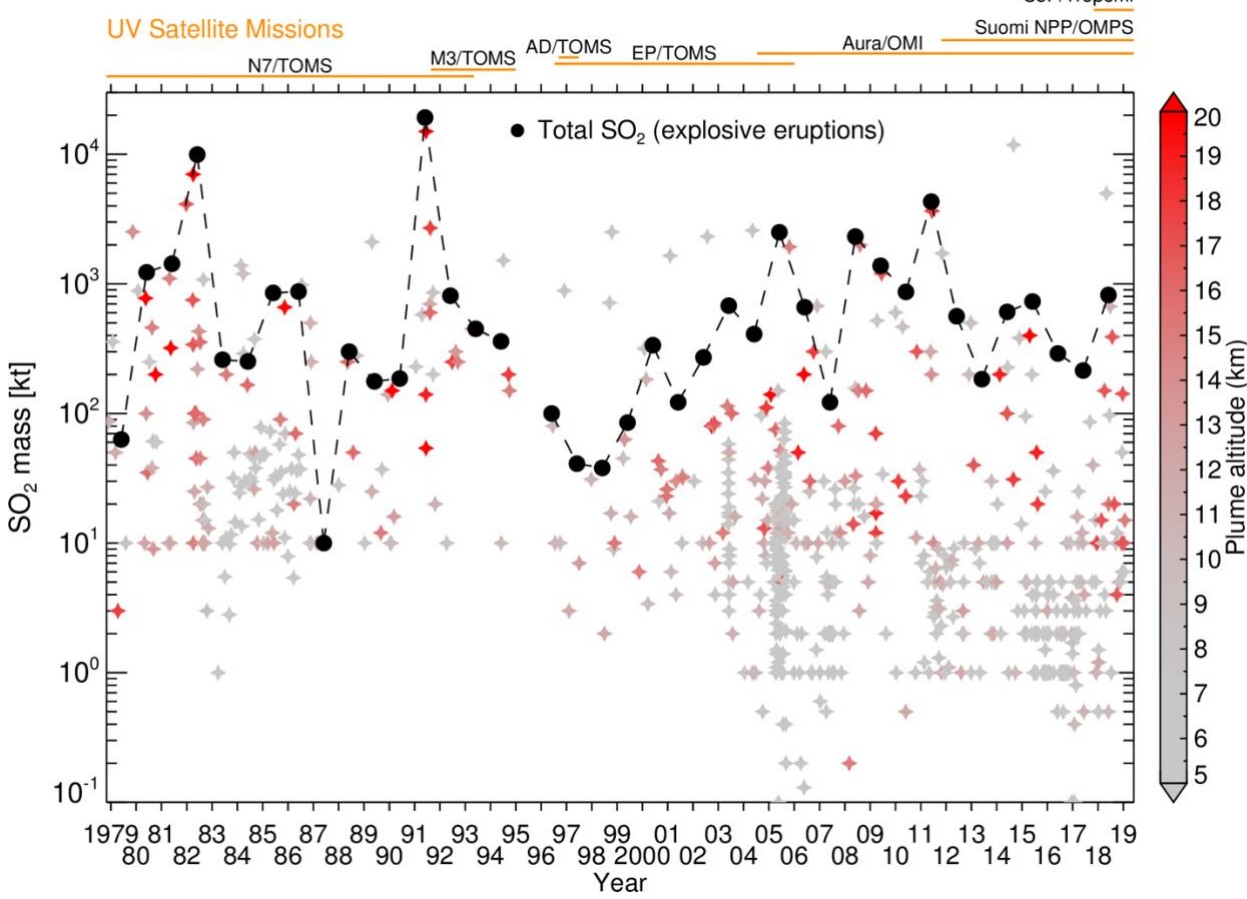

Figure 1. Multi-decadal record of $SO_2$ emissions by volcanic eruptions observed by NASA's fleet of
satellites measuring TOA UV radiances. Eruptions (*star symbols*) are color-coded by estimated plume
altitude, derived from a variety of sources, including Smithsonian Institution Global Volcanism
5 Program volcanic activity reports, volcanic ash advisories, and satellite data. The annual total explosive
volcanic $SO_2$ production (omitting $SO_2$ discharge from effusive eruptions) is shown in black. *Orange
lines* above the plot indicate the operational lifetimes of NASA UV satellite instruments: Nimbus-7
(N7), Meteor-3 (M3), ADEOS (AD), and Earth Probe (EP) TOMS, OMI (currently operational), and
SNPP/OMPS (currently operational), along with the European Sentinel-5P/TROPOMI (currently
10 operational). Data shown in this plot are available from the NASA Goddard Earth Sciences (GES) Data
and Information Services Center (DISC) as a level 4 MEaSUREs (Making Earth System Data Records
for Use in Research Environments) data product (Carn 2019).



## 2 Heritage satellite ozone and SO₂ algorithms

Ozone and $SO_2$ are the two main absorbers in the near UV spectral region between 300 and 340 nm. The relative contributions of each gas to the satellite BUV measurements at the three absorbing TOMS channels used in the retrieval, depend on the spectral structure of the absorption cross sections, which are measured as functions of wavelength and temperature (Bogumil et al., 2003, Daumont et al. 1992). Figure 2 shows the $O_3$ and $SO_2$ cross sections and the $SO_2/O_3$ cross section ratio as a function of wavelength for a spectral UV region spanning the three absorbing channels (317, 331, 340 nm) of TOMS. At the instrument's spectral resolution (~1 nm FWHM) the $SO_2$ molecule is 2.5 times more absorbing than $O_3$ at 317 nm, while $O_3$ is 6 times more absorbing at 331 nm. These differences allow for simultaneous multispectral retrievals of $O_3$ and $SO_2$.

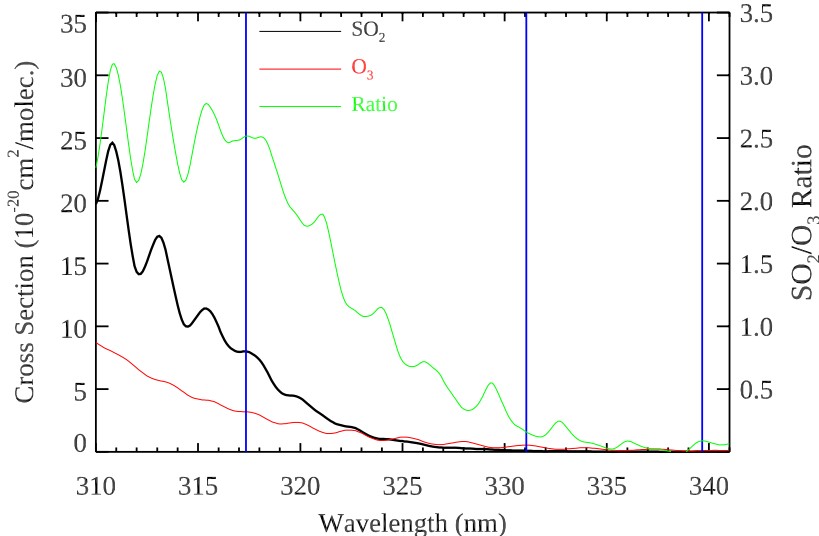

Figure 2. Spectral dependence of laboratory measured $SO_2$ (black) and $O_3$ (red) cross sections between 310-340 nm at TOMS spectral resolution (FWHM~1nm). The $SO_2/O_3$ ratio (green) is shown with the scale on the right axis. The nominal locations of the N7/TOMS absorbing bands (317, 331, 340 nm) are shown by vertical blue lines (blue).

## 2.1 Heritage BUV Ozone algorithms

Dave and Mateer (1967) first proposed a technique to estimate total ozone column from nadir backscatter UV measurements taken in the Huggins ozone absorption band (310-340 nm), assuming no $SO_2$ is present. Their algorithm was inspired by the pioneering Dobson Spectrophotometer which





measures attenuation of solar irradiance by UV wavelength pairs from which total ozone is derived, using the Beer-Lambert law. However, unlike the direct sun technique, radiative transfer calculations show that the top-of-the atmosphere BUV radiances (I) do not follow the Beer-Lambert law. In general, log(I) varies non-linearly with ozone column amount ($\Omega$), and this relationship is sensitive to the shape
of the ozone profile (defined as the ozone density profile normalized to total ozone). To account for this effect Dave and Mateer (1967) proposed constructing a set of lookup tables (LUT) based on standard ozone profiles with different total ozone amounts using ozonesonde and Dobson Umkehr data. Since the shape of the profiles also vary with latitude, they proposed using three sets of profiles for low, mid and high latitudes. These profiles are then used to estimate I, which varies with wavelength ($\lambda$),
observational geometry, surface pressure and surface reflectivity (R). Following the Dobson convention, log(I) is converted to N-value which is defined in Eq. (1) as,

$$N = -100 log_{10}\left(\frac{I}{F}\right) \qquad (1)$$

F is the extra-terrestrial solar irradiance. By linearly interpolating N between total ozone nodes, one forms the N-$\Omega$ curves that are a single valued function of $\Omega$ representative of a given latitude band and
observational geometry. This approach allows $\Omega$ to be estimated by matching the measured N-value to the interpolated N-values.

Over the years several modifications have been introduced to this basic concept. Mateer et al. (1971) proposed a Lambert-equivalent reflectivity (LER) concept to estimate the combined contribution of surface, clouds and aerosols to BUV radiance. In this concept, the scene at the bottom of the
atmosphere is assumed to be a Lambertian reflector whose reflectivity ($R_s$) is derived from the measurements at 380 nm where the ozone and $SO_2$ absorption is negligible. The effective pressure of this reflecting surface is assumed to vary with $R_s$, from a terrain pressure at $R_s < 0.2$ to a fixed cloud pressure 0.4 atm at $R_s > 0.6$, linearly interpolated at intermediate $R_s$. The algorithm assumed that $R_s$, thus derived, did not vary with wavelength. Although in the earlier versions of this algorithm
wavelength pairs (313/331, 318/340) were used to derive $\Omega$, $R_s$ was later derived at 331 nm to minimize errors due to the spectral dependence of $R_s$. This made pairing unnecessary (McPeters et al, TOMS User Guide).

By explicitly modelling the effect of aerosols using a radiative transfer code, Dave (1966) showed that $R_s$ did not vary significantly with wavelength for non-absorbing aerosols, hence they
produced no ozone error. However, for aerosols that might have strong absorption in the UV, he predicted that $R_s$ would decrease at shorter wavelengths, producing an overestimation of ozone. However, since aerosol properties in the UV were not known at that time, no correction for aerosol





absorption was applied until the mid 90s when the effect predicted by Dave (1966) was detected in the Nimbus-7 TOMS data launched in October 1978.

Since the TOMS instrument had three reflectivity channels (331, 340, 380 nm), it was possible to compare the reflectivities derived from them. This comparison showed that $R_s$ increased significantly
with wavelength for moderately thick clouds causing a significant underestimation of $\Omega$ (up to 3%). A modified LER (MLER) concept assuming two Lambertian surfaces, one at the surface and the other at the cloud top was applied to minimize this error (Ahmad et al., 2004).

The most recent version of the TOMS ozone algorithm reverts back to the LER model, but it assumes that clouds are at the surface, which reduces the $R_s$ wavelength dependence (Ahmad et al.,
2004). An estimated ozone amount below the cloud is added to the derived $\Omega$. However, since there are many other reasons for such a dependence including ocean color, non-Lambertian surfaces, such as ocean glint and fogbow, and most importantly the absorbing aerosol effect predicted by Dave (1966), $R_s$ is assumed to vary linearly with $\lambda$; its slope is derived using 340 and 380 nm radiances. This simple omnibus approach works well for most cases, except when the UV absorbing aerosols (smoke, dust and
volcanic ash) are very thick. Such data are flagged in the TOMS ozone algorithm. The new MS_SO2 algorithm is an extension of this algorithm into two dimensions (section 3).

**2.2 Heritage TOMS SO₂ algorithms**

Krueger (1983) was the first to suggest that TOMS could be used to retrieve sulfur dioxide from explosive volcanic events. He correctly interpreted the large positive ozone anomaly observed following
the explosive eruption of El Chichon in 1982 as being due to the $SO_2$ released into the atmosphere during the event. To estimate the $SO_2$ inside the plume region, he separated the $SO_2$ and $O_3$ signals by computing a residual reflectance, estimated as the difference between the unperturbed background ozone outside the plume and the ozone anomaly inside the plume. This early technique for retrieving $SO_2$ from TOMS ozone estimates became known as the residual method. The residual method,
however, failed when the background could not be clearly separated from the ozone anomaly. Krueger subsequently developed the first BUV algorithm that separated the $O_3$ and $SO_2$ radiance contributions, based on an earlier methodology developed by Kerr (1980) to retrieve the $SO_2$ column from the ground with a Brewer spectrophotometer. This method assumed that the TOA backscattered radiation was attenuated by the two absorbing species ($O_3$, $SO_2$), leading to the Eq. (2) describing BUV radiance, I,
for a given wavelength, $\lambda$, corresponding to the TOMS field of view (FoV):

$$I(\lambda) = aF(\lambda) \exp\left[-b\lambda + Sg(\tau_{O_3} + \tau_{SO_2})\right], \tag{2}$$





In Eq. (2), $F$ is the incoming solar flux, $S_g$ is the geometrical optical path (air-mass factor, AMF), and $\tau_{O3}$ and $\tau_{SO2}$ are the absorption optical thicknesses for $O_3$ and $SO_2$, while the coefficients a and b depend on the satellite viewing geometry, cloud/surface reflectance and volcanic ash and sulfate aerosols (Krueger et al., 1995, Krotkov et al., 1997). Equation 2 can be expressed in matrix form, which is then
inverted to obtain estimates for the $SO_2$ and $O_3$ vertical column densities and the dimensionless parameters a and b. This algorithm is generally referred to as the Krueger-Kerr algorithm (Krueger et al., 1995). Krotkov et al. (1997) developed radiative transfer path correction, which explicitly accounted for the $R_s$, ozone and $SO_2$ vertical profiles, replacing the geometrical AMF in Eq. (2). The modified algorithm with empirical background correction has been used off-line on a case-by-case basis for the
past two decades to retrieve $SO_2$ mass tonnage from medium to large explosive eruptions using TOMS BUV measurements (Krueger *et al.,* 2000; Carn *et al*., 2003).

## 3 New MS_SO2 algorithm

       The new discrete wavelength $SO_2$ algorithm (MS_SO2) builds on the heritage of the TOMS total ozone algorithm (section 2.1) but adds sulfur dioxide ($SO_2$) as a second absorber. The BUV
radiance is simulated with the TOMRAD forward vector radiative transfer (RT) model (Dave 1964) from a known viewing geometry by assuming a vertically inhomogeneous, pseudo-spherical Rayleigh scattering atmosphere with standard $O_3$ and a priori $SO_2$ vertical profiles. The underlying reflecting surfaces (land/ocean, clouds and aerosols) are approximated with the simple LER reflecting surface at terrain height pressure (section 2.1). TOMRAD accounts for all orders of polarized Rayleigh scattering
and for the gaseous absorption (i.e., $O_3$ and $SO_2$), using a priori vertical profiles of the gas concentrations and laboratory measured temperature dependent gaseous cross sections (Dave and Mateer, 1967, Bogumil et al., 2003, Daumont et al. 1992). Improvements to the TOMRAD model include corrections for molecular anisotropy (Ahmad and Bhartia, 1995), rotational Raman scattering (Joiner, 1995) and pseudo-spherical corrections to account for changes to the solar and viewing zenith
angles due to the sphericity of the earth.

       Performing on-line radiative transfer calculations for every satellite field-of-view (FoV) greatly increases the time required to process full orbits of data. To improve the computational efficiency of the operational algorithm, N7TOMS-specific look-up-tables (N7TOMS-LUT) were produced off-line using the inputs listed in Table 1 and convolved with the triangular band pass at each of the six Nimbus-7
TOMS wavelengths (FWHM~1 nm). For a given set of inputs, the five BUV radiance parameters, $I_0$, $I_1$, $I_2$, $T$ and $S_b$ are linearly interpolated from the N7TOMS-LUT and the total BUV radiance, $I_c$, , is calculated using Eq. (3):



$$I_c(\lambda, \theta_0, \theta_S, \phi, P, \Omega, \Sigma, R_S) = I_0 + I_1 \cos(\phi) + I_2 \cos(2\phi) + \frac{R_s T}{(1 - R_s s_b)}, \qquad (3)$$

where, $I_c$ is a function of wavelength, solar and satellite zenith viewing angles, relative azimuthal angle, surface pressure, a priori ozone and sulfur dioxide vertical profiles, and the underlying surface reflectivity, respectively. The $I_c$ is converted to N-value, $N_c$, as defined in Eq. (1), and compared with the measured N-value, $N_m$ during operational retrievals.

**Table 1:** Input parameters used in construction of the Nimbus-7 TOMS LUTs

| TOMRAD-based LUT nodes | | |
|---|---|---|
| LUT Node | Number of Nodes | Values |
| Surface Pressure | 2 | 1013.25 and 500 hPa |
| Wavelength | 6 | 312.5, 317, 331, 340, 360 and 380 |
| Total Column Ozone | 21 | 3 low, 8 middle and 11 high latitude bands (Dobson Units, DU) |
| Total Column $SO_2$ | 12 | 0, 5, 10, 50, 100, 150, 200, 250, 350, 450, 550, 650 (DU) |
| solar zenith angle SZA | 10 | 0, 30, 45, 60, 70, 77, 81, 84, 86, 88 |
| satellite zenith angle VZA | 6 | 0, 15, 30, 45, 60, 70 |

The TOMRAD was configured to account for two absorbing trace gases: $O_3$ and $SO_2$. The LUTs include twenty-one total ozone nodes and twelve total $SO_2$ nodes for each of the three assumed $SO_2$ vertical profile shapes. For ozone, the total column amounts correspond to the standard profile shapes, which vary between three latitude bands (see Table 1). For sulfur dioxide, we assumed a Gaussian vertical profile shape, which is determined by two parameters: a center of mass altitude (CMA) and a geometrical standard deviation. The CMA represents the altitude of the peak $SO_2$ concentration. LUTs for $SO_2$ are generated for three different CMAs: 8 km (middle troposphere, TRM), 13 km (Upper Tropospheric/Lower Stratospheric, TRU), and 18 km (lower stratospheric, STL). A constant standard deviation of σ=2 km is assumed for each $SO_2$ profile.

The MS_SO2 algorithm retrieves a four-parameter state vector, **x,** defined below in Eq. (4) as,





$$x = \begin{pmatrix} \Sigma \\ \Omega \\ dRs/d\lambda \\ R_S \end{pmatrix}, \tag{4}$$

where $\Sigma$ is the retrieved total column sulfur dioxide, $\Omega$ is the total column ozone, $dR/d\lambda$ characterizes the $R_s$ spectral dependence, and $R_S$ is the LER at 380nm. The retrieval of sulfur dioxide is carried out in one or two steps described in the next sections, referred to as step 1 and step 2.

## 5    3.1 Step 1 retrieval

The state vector, $x$, is determined iteratively by inverting the Jacobian matrix $\mathbf{K}$ (Eq. 5) at each iteration step:

$$N = \mathbf{K}x, \tag{5}$$

where $N$ represents the measured vector of N-values for the four TOMS channels at 317, 331, 340 and
380 nm, and $\mathbf{K}$ represents a 4 x 4 Jacobian matrix derived from the LUTs. The matrix elements are defined as:

$$K_{i,j} = \frac{\partial N_{c\,i}}{\partial x_j}, \quad i,j = 1,4 \tag{6}$$

where $N_{c,i}$ is the forward model calculated N-value at wavelength $i$.

Our step 1 inversion starts with an initial state vector $\mathbf{x_0}$, consisting of first guesses for $\Sigma_0$, $\Omega_0$,
and $dR_s/d\lambda$ shown in Table 2. Rs is computed analytically using the measured BUV radiance at 380 nm, before beginning the iteration (see Supplement, Eq. S2). Note that since the $O_3$ and $SO_2$ cross-sections are negligible at 380 nm, the Rs and $\partial N_{380}/\partial R_s$ do not change with the iterations (*i.e.,* $dR_s = 0$).

Equation (5) is solved iteratively by zeroing the residuals, $dN=N_{meas}-N_c$, and re-computing the Jacobians at each iteration step for the four used spectral channels. The state vector is then adjusted
after each iteration, $\mathbf{x} = \mathbf{x_0} + \mathbf{dx}_k$, until it converges on a solution (*i.e.,* $\mathbf{dx}_k \sim 0$), as described in Eq. (7) below:

$$dN_k = N_m - N_{c,k-1} = \mathbf{K}_{k-1}dx_k \tag{7a}$$

$$dx_k = x_k - x_{k-1} = \mathbf{K}_{k-1}^{-1}dN_k \tag{7b}$$

Since $O_3$ and $SO_2$ exhibit small absorption at 340 nm, a non-zero R spectral slope (i.e., dR/d$\lambda\neq$0)
accounts for the radiative effects of aerosols. As indicated in Table 2, the algorithm initially assumes no R-lambda dependence (i.e., dR/d$\lambda$=0), however, absorbing and non-absorbing aerosols can cause dR/d$\lambda\neq$0.



**Table 2**: State Vector

| Retrieved Parameter | Wavelength (nm) | Symbol | First Guess[*] |
|---|---|---|---|
| Total Column $SO_2$ | 317 | $\Sigma$ | $\Sigma_0 = 0$ |
| Total Column $O_3$ | 331 | $\Omega$ | $\Omega_0$ |
| Spectral Reflectivity Dependence | 340 | $dR/d_\lambda$ | $dR_0/d_\lambda = 0$ |
| Reflectivity | 380 | $R_S$ | N/A |

*$\Omega_0$ is a climatological value corresponding to one of three latitude bands

The algorithm uses the retrieved spectral slope, defined in Eq (8), to update the calculated LERs after each iteration:

$$R_j = R_S + \frac{\partial R}{\partial \lambda}\left(\lambda_j - \lambda_R\right), \quad j = 1, 4 \tag{8}$$

where $\lambda_j = 312, 317, 331, 340$ nm and $\lambda_R = 380$nm. When $SO_2$ or aerosol loading is high, non-linear
effects can cause the $SO_2$ Jacobian in Eq. (6) to become very small causing R(j) to deviate significantly
from Eq. (8), especially at the shortest TOMS wavelength, 312nm. The non-linear spectral dependence
of R(j) produces systematic errors in the retrieved state vector. For this reason, we do not use the 312
nm channel in the retrievals, but the final residual $dN_{312}$ can be used as a diagnostic of the spectral non-
linearity of the scene. A step 2 empirical procedure, described in the next section, was developed to
correct for the retrieval bias resulting from these errors.

**3.2 Step 2 retrieval**

The MS_SO2 forward model accounts for $O_3$ and $SO_2$ absorption and linear spectral changes in
$R_s$ due to the presences of aerosols. The algorithm, however, does not explicitly characterize the
absorption and scattering effects of volcanic ash (absorbing) and sulfate (non-absorbing) aerosols. The
retrieval errors in $\Sigma$ and $\Omega$ caused by volcanic ash during the first days after an explosive eruption can
be significant in the case of major volcanic eruptions like Pinatubo and El Chichon (Krueger et al.,
1995; Krotkov et al., 1997). A step 2 procedure was developed primarily to handle explosive eruptions
(VEI > 3), in which large $\Omega$ anomalies are identified to occur in conjunction with high ash
concentrations. In step 2, a corrected total ozone $\Omega_{cor}$ inside the $SO_2$ cloud is interpolated using the





retrieved $\Omega$ outside the plume along the orbit for each cross-track position. Even if ozone destroying halogens are present, such effects can still be considered negligible over the relatively short time periods that SO₂ concentrations are high enough to affect TOMS observations.

In deciding whether to apply Step 2, the algorithm considers the retrieved $\Sigma$, $\Omega$ and Aerosol
Index (AI) in Step 1. The AI is estimated from Eq. (9) from the Step 1 parameter dR/dλ and the calculated Jacobian dN/dR at 340 nm:

$$AI = \frac{\partial N_{340}}{\partial R}\frac{\partial R}{\partial \lambda}(\lambda_{340} - \lambda_{380}) = -40 \cdot \frac{\partial N_{340}}{\partial R}\frac{\partial R}{\partial \lambda}. \qquad (9)$$

Positive AI (dR/dλ>0) identifies spatial regions affected by absorbing aerosols (dust, smoke, ash). The step 2 selection criteria first select FoVs where either SO₂ > 15 DU (inside the plume) or AI > 6. The
additional AI criterion allows for the selection of FoVs around the edges of the cloud, where the SO₂ can be less than 15 DU due to high aerosol concentrations. In this case, it is assumed that the step 1 SO₂ may have been underestimated due to the ozone error caused by high aerosol concentrations (in these cases, the SO₂ retrieved in step 2 may still not exceed 15 DU, and therefore would be excluded from the plume in subsequent mass calculations). We describe the methodology for interpolating $\Omega_{cor}$ in
equations S3-S5 of the supplement. A second retrieval of SO₂ and dR/dλ is then performed by inversion using the measured 317 and 340 nm radiances while treating the ozone $\Omega_{cor}$ as a constant. This constraint on the ozone bounds the SO₂ Jacobians computed from the forward model LUTs. The operational MS_SO2 file includes a step 2 algorithm flag (not applied = 0, applied = 1).

To illustrate the effects of the step 2 procedure, we consider the 1982 explosive eruption of El
Chichon, which emitted ~7 Tg SO₂, the second largest observed in the satellite era (Fig.1). Figure 3 shows the retrieved AI map during TOMS overpass of the volcano on April 4, 1982, while it was still erupting. High AI values exceeding a value of 10 correspond to high biased step 1 ozone values (Fig. 4a) and underestimated $\Sigma$ values (Fig 4c). Figure 4b shows the step 2 corrected $\Omega_{cor}$, which is consistent with the $\Omega$ field outside of the volcanic cloud. Figure 4d shows the step 2 $\Sigma$, which is much higher than
retrieved in step 1. As can be seen in this particular example, the step 2 correction can significantly increase the SO₂ mass. In this case, the SO₂ mass increased from 2475 kilotons (step 1 to 3637 kilotons (step 2). Peak $\Sigma$ values increased from 396 DU to 549 DU in the aerosol affected region. The biases, d$\Omega$ and d$\Sigma$, for this case are shown in Fig. S1 of the Supplement. Step 2 was developed primarily to handle extreme eruptions (VEI > 3), such as El Chichon and Pinatubo, where large $\Omega$ anomalies sometimes
occur in conjunction with high ash concentrations. In practice step 2 corrections tend to be small (or none at all) for most  moderate eruptions detected during the observation period covered by Nimbus 7 TOMS.



The step 2 $\Omega$ values inside the volcanic cloud shown in Fig. 4b appear to be fairly consistent with the surrounding ozone field, but there still exist a few remaining high $\Omega$ values in the boundary of the plume, which were not selected for step 2 (Fig. 4b). These pixels were not corrected because the threshold criteria were not met, so that $\Sigma$ may be underestimated. However, their contribution to the total $SO_2$ cloud mass is insignificant.

Step 2 follows a methodology similar to the original residual method developed by Krueger (1983), which separated the $O_3$ and $SO_2$ contributions by subtracting the measured BUV reflectance in the unperturbed region from the BUV radiance anomaly associated with the $SO_2$ cloud. The MS_SO2 algorithm corrects the ozone inside the plume by removing the ozone anomaly. Our Step 2 procedure is typically only applied when the ash loading (i.e., high AI) causes the reflectivity dependence to become non-linear, as the forward model does not explicitly account for volcanic aerosol absorption. This scenario typically lasts for about 1-3 days following a major explosive eruption, during which total retrieved $SO_2$ mass is likely to be underestimated, and in some cases, could even increase with time due to ash and ice fallout and plume dispersion. For such extreme cases we recommend estimating $SO_2$ to sulfate conversion e-folding life-time using weeks' worth of measurements of the total $SO_2$ cloud daily mass and extrapolating it back in time to estimate total $SO_2$ mass emitted on eruption day. This "day one" time extrapolated $SO_2$ mass is typically larger than retrieved on days immediately following the eruption (Krotkov et al., 2010).

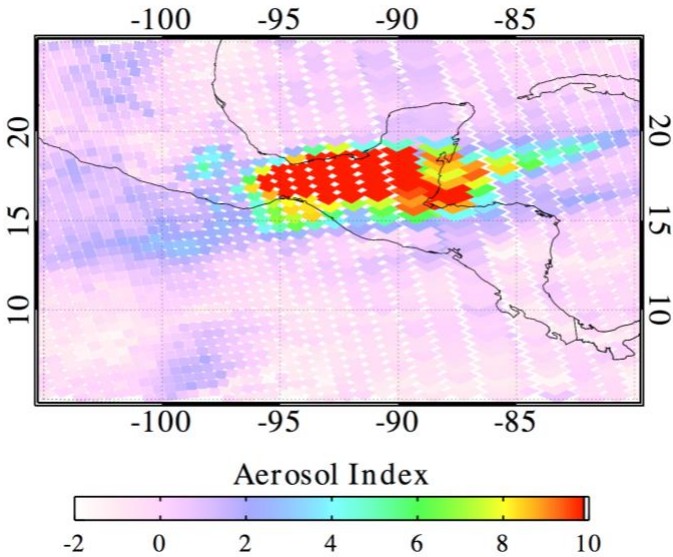

**Figure 3.** Aerosol Index for the El Chichon eruption on April 4, 1982, computed from retrieved $dR_\lambda$.



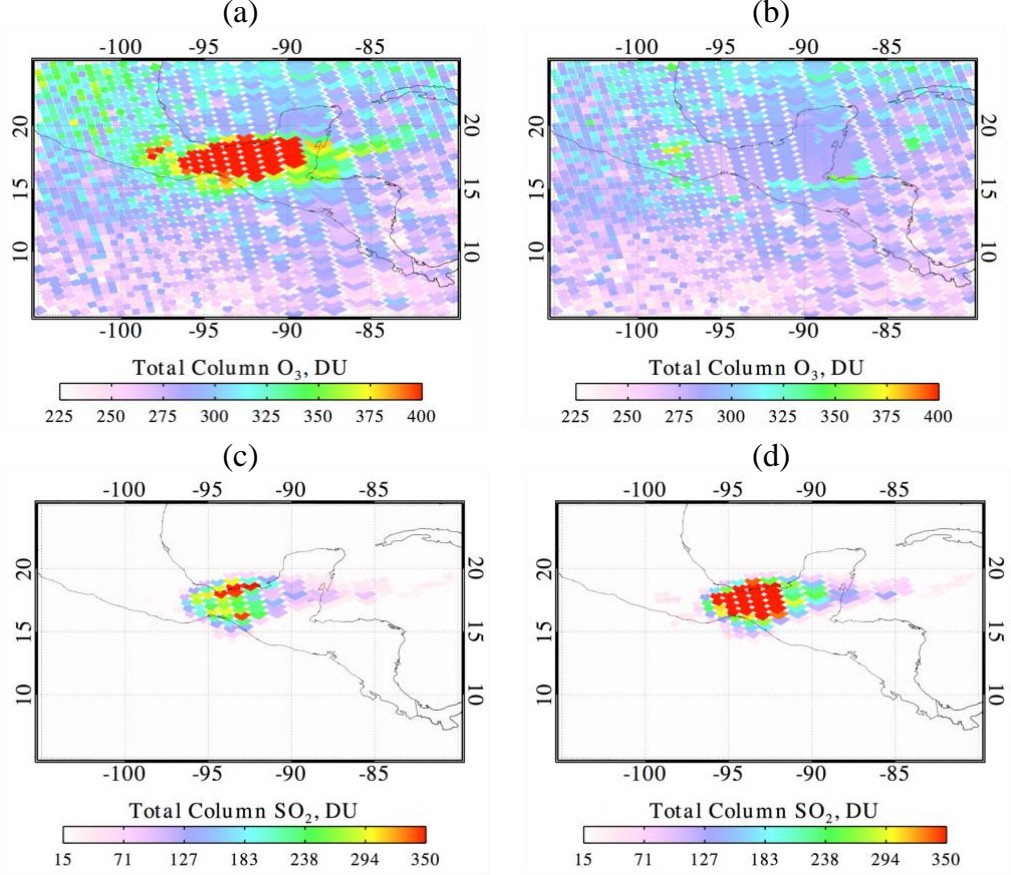

**Figure 4.** MS_SO2 imagery showing a) Step 1 total column $O_3$, b) Step 2 total column $O_3$ c) Step 1 total column $SO_2$ and d) Step 2 total column $SO_2$ for the case of El Chichon eruption on April 4, 1982.

### 3.3 Soft Calibration: N-value bias correction

We assume that the background sulfur dioxide is below TOMS detection limit in regions of the atmosphere far away from any $SO_2$ sources (e.g., volcanic, anthropogenic). Random errors associated with the retrieval process, however, are normally distributed around zero. We expect that the true

10  volcanic $SO_2$, $\Sigma_{true}$, and the mean of the background $SO_2$ distribution, $\langle \Sigma \rangle_{clean}$, are zero:

$$\Sigma_{true} = \langle \Sigma \rangle_{clean} = 0 \qquad (10)$$

An examination of the measured $\Sigma$ distribution in clean regions of the Central Pacific Ocean found a positive bias of about 3 DU (i.e., $\langle \Sigma \rangle_{clean} \sim$ 3 DU, Fig. 5). We correct for the bias by applying a



small fixed N-value adjustment at 340 nm as a function of the swath position (Figure 6). Figure 5b shows the effect of this adjustment on the step1 background $\Sigma$ distribution (Fig. 5a). Figure 5c displays daily global $SO_2$ background map, and figure 5d shows the $SO_2$ difference map (after – before the correction). The details of this procedure are described in S3.3 of the supplement.

Figure 5. Probability density of function of $SO_2$ background a) before applying 340 N-value correction and b) after applying correction, c) global daily $SO_2$ map before applying 340 N-value correction, d) the difference (after correction – before correction) map.



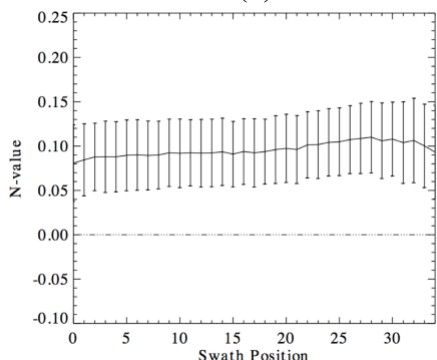

Figure 6. Mean 340 nm N-value adjustment with error bars computed from a sample of 90 orbits, shown as a function of the TOMS swath position. The error bars are estimated from the standard deviation computed from the 90 orbits. The correction is subtracted from the measured $N_{340}$-value.

## 4. Error analysis

### 4.1. Random errors and SO$_2$ detection limit.

The random errors in the MS_SO2 retrieval were estimated from the standard deviation in the SO$_2$ from a large data sample that included 90 central Pacific orbits, spanning a ten-year period between 1980 and 1990. Data were restricted to $\Sigma$ values between -20 and 20 DU (Fig. 7a). Standard deviations were then computed as a function of the TOMS swath position as shown in Fig. 7b. Figure 7b can be used to characterize the SO$_2$ detection limits for TOMS. In this section, we compare the TOMS error distribution with the UV Ozone Mapping Profile Suite Nadir Mapper (OMPS-NM), a hyperspectral UV instrument on board the Suomi National Polar-orbiting Partnership (NPP) and NOAA 20 satellites. For this comparison, we selected one month of NPP/OMPS spectral data (central Pacific) and applied the MS_SO2 algorithm using the same four wavelength bands on TOMS (Table 2), which were first convolved with the TOMS bandpass function.

Figure 7b shows that TOMS retrieval noise depends on the swath position, varying from ~ 6 DU at nadir to ~4 DU at higher viewing angles, while OMPS is 2-3 times smaller (~2 DU) and is relatively independent of the cross-track position (Figure 7). Using the MS_SO2 algorithm, we subsequently estimate the SO2 detection limit for TOMS and OMPS-NM to be about 15 DU and 6 DU (~99% confidence level), respectively. We note that applying the Principal Component Algorithm (PCA) (Li et al., 2013) to all the 100-200 wavelengths available from OMPS-NM hyperspectral measurements, the noise spectrum is reduced by an order of magnitude to ~0.2-0.5DU (Zhang et al., 2017), allowing detection of large anthropogenic points sources (emissions more than ~80 kt yr$^{-1}$) (Fioletov et al., 2016).



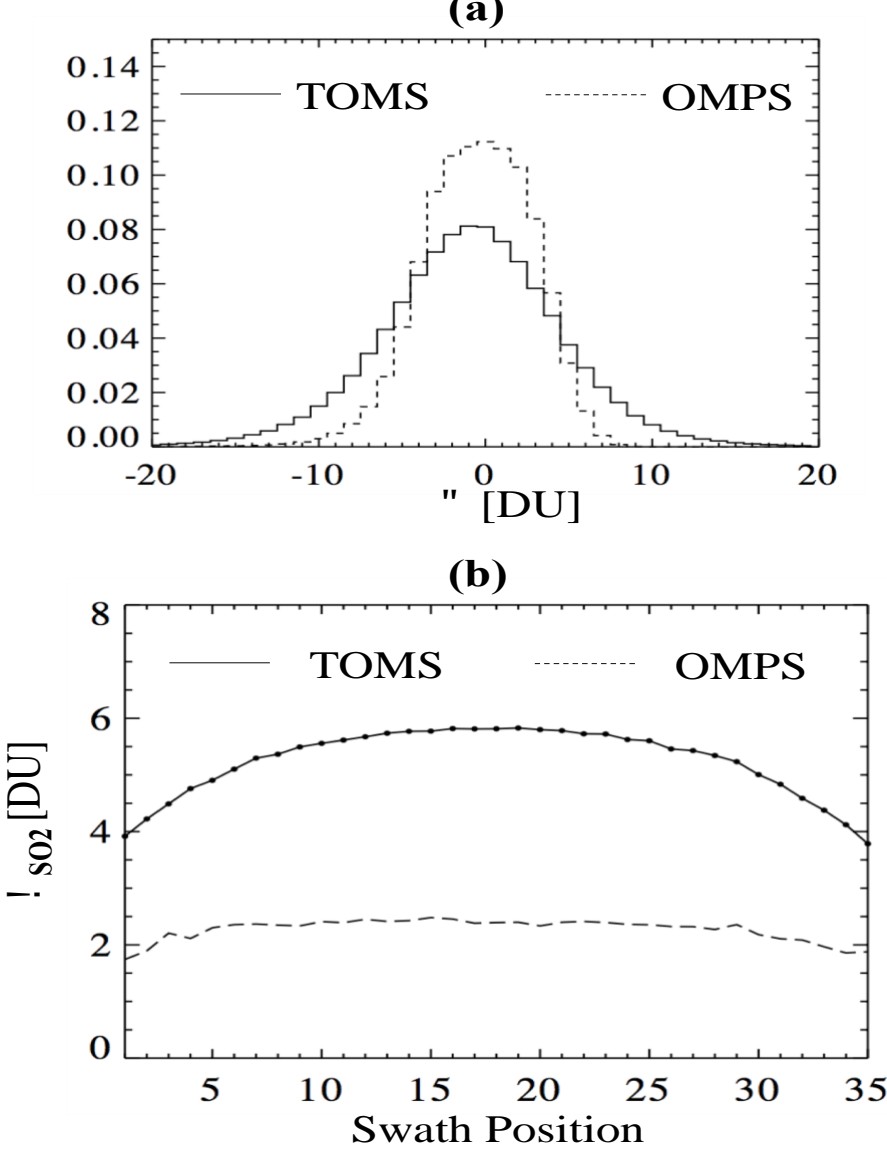

Figure 7. (a) Background retrieved SO₂ noise histogram for Nimbus-7 TOMS (solid line) and current operational hyperspectral Ozone Mapping Profiling Suite (OMPS) nadir mapper (dashed line) for all swath positions; (b) Noise standard deviations of background SO₂ retrievals as a function of the swath position. OMPS noise is factor of 2-3 less than TOMS and less dependent on cross-track position.



## 4.2 Systematic errors in volcanic SO₂ plumes

In this section, we evaluate systematic errors of the MS_SO2 retrievals of volcanic $SO_2$. The two most significant errors are caused by volcanic aerosols (ash and sulfate) and incorrect assumptions regarding the $SO_2$ profile, namely the plume height. The radiance LUTs used by the algorithm account for ozone and $SO_2$ absorption, but do not account for the absorption and scattering by aerosols. The ash errors can be significant during the first couple days after the initial eruption phase (Rose et al., 2003, Guo et al., 2004). The pre-computed radiance tables used by MS_SO2 assume an $SO_2$ column amount based on an a priori CMA and standard deviation (Section 3). An incorrect CMA assumption can cause significant $SO_2$ errors that vary with viewing geometry, ozone and $SO_2$ column amounts. We characterize these errors by applying the MS_SO2 algorithm to synthetic radiances.

### 4.2.1 Uncertainties due to SO₂ plume height

To understand retrieval errors in MS_SO2 algorithm due to assumed a priori $SO_2$ profiles, we calculated column $SO_2$ Jacobians $\partial N/\partial \Omega_{SO_2}$ using the VLIDORT radiative transfer code (Spurr and Christi 2019) for the typical observational conditions in the tropics, mid latitudes, and high latitudes. Figure 8 shows $\partial N/\partial \Omega_{SO_2}$ at 317 nm, for different $SO_2$ amounts, nadir angles and scene reflectance as function of the assumed $SO_2$ height (center of mass altitude, CMA). The Jacobians generally increase with the CMA, meaning that satellite BUV measurements are more sensitive to $SO_2$ at higher altitudes. This means that the MS_SO2 algorithm will overestimate (underestimate) the $SO_2$ column amount, if the CMA of the a priori profile is lower (higher) than that of the actual $SO_2$ profile. On the other hand, the sensitivity of $SO_2$ Jacobians with respect to CMA is affected by several factors, particularly $SO_2$ column amounts, geometry (solar zenith angle and viewing zenith angle), the reflectivity of the underlying surface (Rs), and the CMA itself. In general, the sensitivity of $SO_2$ Jacobians to CMA is greater for $SO_2$ plumes with large $SO_2$ loading (e.g., 300 DU *vs.* 50 DU), at relatively low altitudes (e.g., CMA of 13 km *vs.* 18 km), and for lower reflectivity (e.g., Rs of 0.05 *vs.* 0.50) or are near the edge of the swath (e.g., VZA of 60° *vs.* 0°). For calculations assuming typical mid- and high-latitude conditions, we found similar sensitivities of $SO_2$ Jacobians to CMA. From these calculations, we can estimate the errors in the $SO_2$ Jacobians at 317 nm, assuming that the standard a priori profiles used in MS_SO2 retrievals (CMA: 13 and 18 km) have a ±2 km error in CMA. The results for the tropics, mid latitudes, and high latitudes are summarized in the supplement Tables S1, S2, and S3, respectively. As shown in the tables, for $SO_2$ plumes from relatively moderate eruptions (~50 DU), the relative errors in $SO_2$ Jacobians due to the error in the CMA are mostly within ±10%. But for plumes with large $SO_2$



loading (~200-300 DU) from explosive eruptions such as Pinatubo, the relative error in $SO_2$ Jacobians may reach as high as 30% for pixels near the edge of swath that have low reflectivity. Additionally, for pixels with the same reflectivity and VZA, the relative errors due to $SO_2$ height are greater for mid- and high-latitude eruptions than for tropical eruptions.

5        To quantify the retrieval errors due to inaccuracies in the a priori profiles, we used the top-of-the-atmosphere synthetic radiance data generated by VLIDORT, as input to the MS_SO2 algorithm. The retrieved $SO_2$ and $O_3$ column amounts were compared with assumed in VLIDORT calculations (Tables S4-S7 in the supplement). As shown in the tables, for $SO_2$ plumes with a modest loading (~50 DU), the relative errors in $SO_2$ column amounts, due to a 2-km error in the *a priori* profile are typically 10% or less, whereas the relative errors in $O_3$ are within 1%. For plumes with large $SO_2$ loadings (200-300 DU), the errors in $SO_2$ amounts due to a 2-km bias in the *a priori* profile are typically 5-15%, but can reach as high as 30-40% for high latitude plumes with large SZA and VZA. For extreme conditions at high latitudes (Table S5, 13 km *a priori* profile *vs.* 15 km actual profile, $SO_2$=300 DU, VZA=60°), the MS_SO2 algorithm failed to converge after 20 iterations, due to a signal saturation caused by strong absorption at 317 nm. In these relatively rare cases, it is beneficial to use longer wavelengths (e.g., > 320 nm) for $SO_2$ retrievals (e.g., Li et al., 2017; Theys et al., 2015), which are available from the current hyperspectral instruments such as OMI and OMPS, but not TOMS.

        We also calculated the diagnostic residual at 312 nm ($res_{312} = N_{meas} - N_{calc}$), defined here as the difference between the "measured" synthetic $N_{meas}$ and the $N_{calc}$ at 312nm using MS_SO2 retrieved ozone and $SO_2$ column amounts. Note that the 312 nm channel was not used in the MS_SO2 algorithm, and the residuals at other wavelengths are essentially zero since we are retrieving four parameters from four wavelengths. As shown in the Supplement Tables S4-S7, a positive bias in the $SO_2$ height (CMA too high as compared with the actual profile) leads to negative residuals at 312 nm, whereas a negative bias in *a priori* profile (CMA too low) causes positive residuals. The residuals are generally within 1-2 $N$ value (2% -5% error in radiance) for $SO_2$ column amounts of 50-100 DU, but can reach 3-7 $N$ value (6% -15%) for large $SO_2$ amounts of 200-300 DU. While the 312 nm channel may potentially be used to retrieve $SO_2$ plume height for large volcanic eruptions, it is also strongly affected by volcanic aerosols as demonstrated in the next section.



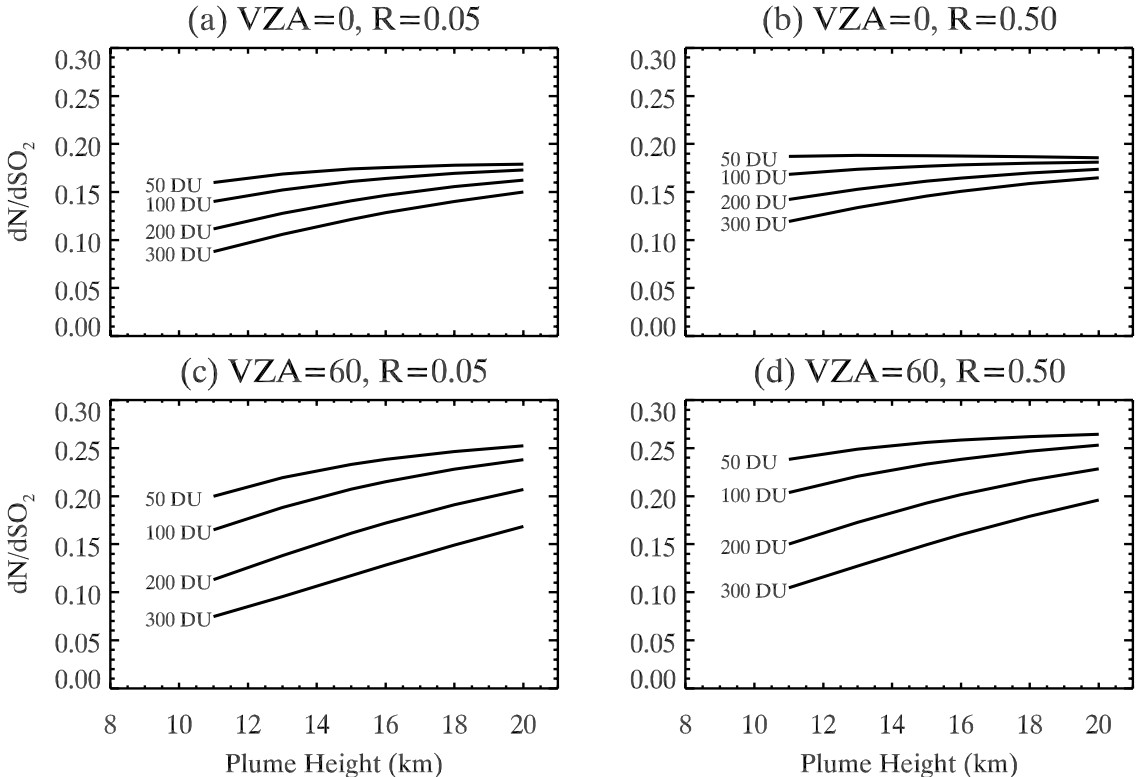

Figure 8. VLIDORT calculated $SO_2$ column Jacobians ($\partial N/\partial SO_2$) at 317 nm for typical conditions in the tropics (SZA=10°, RAZ=90°, $O_3$ = 275 DU) but different $SO_2$ column amounts (50, 100, 200, and 300 DU), center mass altitudes (11-20 km), viewing zenith angle (VZA = 0° and 60°), and SLER (R = 0.05 and 0.50). For these calculations, Gaussian $SO_2$ profiles with the same standard deviation (2 km) were assumed.

**4.2.2 Ash and sulfate aerosol effects on MS_SO2 retrievals**

To test the sensitivity of the MS_SO2 algorithm to ash and sulfate aerosols, an Observing System Simulation Experiment (OSSE) was conducted. The experiment used the GEOS-5 earth system model (*Molod et al., 2012, Buchard et al., 2017, Colarco, et al, 2012*), coupled with the online Goddard Chemistry Aerosol and Radiation (GOCART) (*Chin et al., 2000; Colarco et al., 2010, and references therein*) and the Community Aerosol and Radiation Model for Atmospheres (CARMA) (*Toon et al. 1988; Ackerman et al. 1995; Colarco et al., 2014*). In this experiment, we considered three separate cases for a Pinatubo-like eruption scenario: 1) 12 Mt of $SO_2$ and no aerosols; 2) 12 Mt of $SO_2$ plus 4 MT of sulfate aerosols (as reported by Guo et al. ,2004) and 3) 12 Mt of $SO_2$, 4 Mt of sulfate aerosols



plus 5 Mt of ash uniformly distributed between 18 km and 22km above the location of Pinatubo volcano, on June 15, 1991, from 06:00 – 15:00 UTC.

The GEOS-5 simulated 4D profiles of ozone, $SO_2$, sulfate aerosols, and volcanic ash were used as input to a VLIDORT RT model (Spurr and Christi 2019). The model generated synthetic radiances at
5   317, 331, 340 and 380 nm TOMS bands, using the actual SNPP/OMPS-NM viewing geometry, and assuming cloud-free conditions. The synthetic radiances produced by the VLIDORT were used as input to the MS_SO2 algorithm to generate "retrieved" columns of ozone and $SO_2$. We note that MS_SO2 algorithm uses LUTs produced using a different TOMRAD RT model (Dave 1964).

Figure 9 compares retrieved versus true $SO_2$ column amounts for the three cases considered. The
10   retrieval bias is inferred from the differences between the model $SO_2$ input and the $SO_2$ retrieved by MS_SO2, using the radiances from the model run. The no aerosol case confirms an unbiased $SO_2$ retrievals for $SO_2$ column amounts less than ~150 DU and small positive bias for larger $SO_2$ amounts. For aerosol cases where sulfates and ash were included in the simulation, we observe a negative bias for $SO_2$ column amounts exceeding ~100 DU. These negative biases (retrieval minus input) are expected as
15   the MS_SO2 forward model does not explicitly account for volcanic aerosols. This OSSE experiment shows the effects of heavy aerosol loading on the retrieval, but also increases confidence in MS_SO2 retrievals between 15-100 DU, under nominal conditions, even in the presence of high aerosol concentrations.

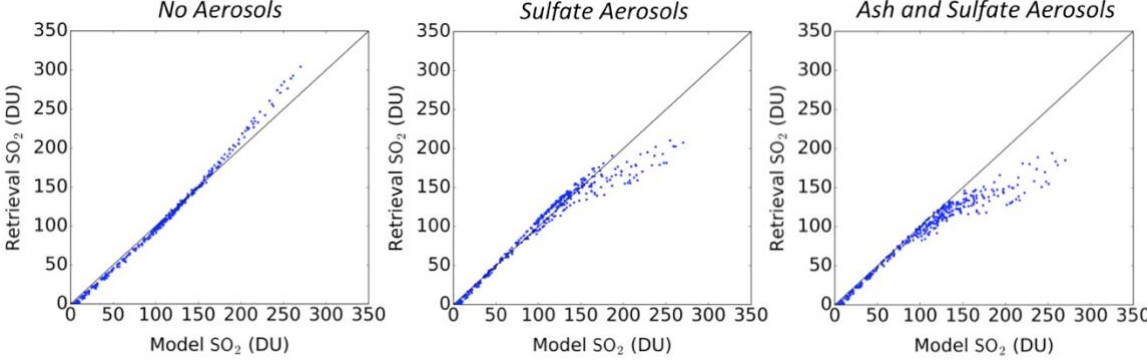

**Figure 9.** Comparison of the OMPS retrieval $SO_2$ against the GEOS-5 model $SO_2$. The TOA radiances for the OMPS retrieval were generated assuming no aerosol (left panel), only non-absorbing sulfate aerosols (center panel), and both ash and sulfate aerosols (right panel).



## 5 Comparison with PCA SO₂ retrievals

We validated the algorithm by directly comparing MS_SO2 retrievals with the independent principal component analysis (PCA) SO₂ algorithm adapted to the TOMS 6 spectral channels. In the PCA approach (Li et al., 2013; 2017), a set of principal components (PCs) is first extracted from the measured radiances using a PCA technique and ranked in descending order according to the spectral variance they each explain. If derived from SO₂-free areas, these PCs represent geophysical processes (*e.g.*, ozone absorption) and measurement details (*e.g.*, wavelength shift) that are unrelated to SO₂, but may interfere with SO₂ retrievals. Next, we fit the first $n_v$ (non-SO₂) PCs and the SO₂ Jacobians ($\partial N / \partial \Omega_{SO_2}$) to the measured radiances (in $N$ value) described in Eq. (11). This allows us to simultaneously estimate the coefficients of the PCs ($\boldsymbol{\omega}$) and SO₂ column amount ($\Omega_{SO2}$), and helps to minimize the impacts of various interfering processes:

$$N(\omega, SO_2) = \sum_{i=1}^{n_v} \omega_i v_i + \Omega_{SO_2} \frac{\partial N}{\partial SO_2} N. \qquad (11)$$

A more detailed introduction to the PCA SO₂ retrieval technique for hyperspectral instruments such as the Ozone Monitoring Instrument (OMI) and the Ozone Mapping and Profiler Suite Nadir Mapper (OMPS-NM) can be found elsewhere (e.g., Li et al., 2013, 2017; Zhang et al., 2017).

For this comparison we adapt the PCA to discrete wavelength of N7/TOMS. The Nimbus-7 TOMS PCA SO₂ algorithm is similar to the OMI and OMPS-NM version in terms of its overall structure but differs in some implementation details. Specifically, unlike the OMI/OMPS volcanic SO₂ retrievals that use a dynamic spectral fitting window (Li et al., 2017), the TOMS PCA SO₂ algorithm uses all six wavelengths available from TOMS in fitting. Also due to the small number of wavelengths, in the TOMS PCA SO₂ algorithm, we always use $n_v = 5$ PCs in Eq. (11), less than the number of PCs used for OMI ($n_v \leq 20$) or OMPS ($n_v \leq 15$). For OMI and OMPS retrievals, SLER is derived at three wavelengths (342, 354, and 367 nm) and extrapolated to other wavelengths using a second-degree polynomial function fitted to these three wavelengths. As for TOMS, SLER is determined at 340 and 380 nm and extrapolated linearly. Additionally, while the Jacobians lookup tables are constructed using the VLIDORT radiative transfer code (Spurr and Christi 2019) for both OMI/OMPS and Nimbus-7 TOMS, different, instrument-specific slit functions are used to band-pass the SO₂ Jacobians from the lookup tables.

We compared retrievals from the two algorithms for the first six days of Mount Pinatubo eruption (June 16-21, 1991). The Pinatubo case provides a large sample of FoVs spanning a broad range of SO₂ amounts from 15 DU (minimum threshold) to over 400 DU. In this test of the algorithm, MS_SO2 and PCA retrievals were generated assuming a CMA =18 km.




## 5.1 June 1991 eruption of Mount Pinatubo

Mount Pinatubo is a large stratovolcano located at 15°08' N,120°21'E in western Luzon, Philippines, that erupted explosively on June 15, 1991, following weeks of precursory activity. TOMS $SO_2$ imagery on June 15 shows a narrow, elongated $SO_2$-ash plume extending to the west from the location of the volcano. On the following day TOMS measured a massive $SO_2$ plume to the west of the volcano (Bluth et al., 1992). TOMS continued tracking the daily evolution of the Pinatubo volcanic cloud as it encircled the earth over a period of about 22 days. Previous estimates of the Pinatubo $SO_2$ height (CMA) range between 18 and 25 km (Self et al., 1994; Guo et al., 2004).

Figure 10 shows TOMS daily $SO_2$ maps produced with the MS_SO2 and the PCA algorithms for the six-day period from June 16 to June 21. Corresponding Ash Index (AI) imagery from MS_SO2 are shown in Figure 11. The $SO_2$ and AI maps for June 16 show a large $SO_2$ and ash cloud propagating to the west. The AI values range from 4 to above ~10 across the cloud. The AI values decreased over the following days due to wind advection and wet deposition (Guo et al., 2004). As the $SO_2$ cloud area continues to expand, total $SO_2$ mass remain high, while $SO_2$ peak values decrease, which is expected from the cloud dispersion. The MS_SO2 and the PCA imagery show excellent qualitative agreement in the total cloud area and internal $SO_2$ plume structure, as inferred from the $SO_2$ gradients across the peak regions of the cloud. Note that for June 16-19, part of the observed cloud is missing due to a known mechanical problem with the TOMS instrument. These missing regions can be clearly identified in the imagery.

Figure 12 shows a scatterplot comparing the MS_SO2 and PCA retrievals for the 6-day time series, which included over 7000 matching FoVs. These results show the retrievals are in close quantitative agreement, with a correlation of 0.993 and a unit slope. Since the two algorithms apply fundamentally different approaches to retrieving $SO_2$, this level of agreement is impressive considered over such a broad range of values.

We further compared quantitative retrievals of the total $SO_2$ cloud mass, peak $SO_2$ values and the cloud area. For this comparison, we also considered results from the Krueger-Kerr algorithm (KK), based on the published results of Guo et al. (2004). Table 3 displays daily estimates of the $SO_2$ cloud mass and peak $SO_2$ amounts for the MS_SO2, PCA and KK algorithms for the six-day period. Guo et al. (2004) applied a modified version of the KK algorithm that assumes a radiative transfer air mass factor (AMF), which accounts for the a priori ozone and $SO_2$ absorption profiles (Krotkov et al., 1997). The early $SO_2$ mass estimates by Bluth (1992) derived from Pinatubo eruption assumed a simplified geometrical AMF. Also note that Guo et al. (2004) interpolated across the missing data regions of the plume on June 16, June 18 and June 19 using a Punctual Kriging statistical analysis. Here, we did not correct for the missing data. The three algorithms are in good overall agreement for the period from



June 17 to June 21, with the differences within 10% compared to MS_SO2. The most significant differences between the three algorithms are observed on June 16 under conditions of heavy ash loading. The KK SO$_2$ mass exceeded MS_SO2 by over ~20%, while the MS_SO2 and the PCA mass retrievals differ by just 2%. Some of the difference between KK and the other two algorithms can be
attributed to the fact that the Guo et al. (2004) estimates include contribution from the missing data region at the northern boundary of the plume (compare SO$_2$ and aerosol imagery), but this contribution does not nearly account for the total difference in Table 3.

The differences can be explained by considering how each algorithm are affected by volcanic aerosols. The MS_SO2 accounts for ash implicitly by retrieving the LER spectral dependence at 340
nm, which is then used iteratively to correct the LER at the two absorbing channels. As explained in Sec. 3.2, absorbing aerosols cause possible apparent retrieved ozone increase, which decrease Σ. The KK algorithm (Krueger et al., 1995) also accounts for ash implicitly by retrieving two linear spectral parameters a and b that adjust to match $N_{calc}$ and $N_{meas}$ for all spectral channels, including the shortest 312nm.  The KK radiative path LUTs are based on TOMRAD calculations that do not explicitly
account for ash (Krotkov et al., 1997). Krueger et al. (1995) estimated that ash aerosols can cause errors in the retrieval up to +30%, depending on the ash size distribution. The measurements driven PCA algorithm, in contrast, accounts for ash empirically by separation and ordering of the measured spectral principal vectors. The differences between MS_SO2 and KK on June 16 and June 17 can be partly ascribed to the effects of aerosols on the retrievals.

By June 18, the ash and SO$_2$ clouds have mostly separated, though, aerosol indices over 4 are still observed in some regions of the plume. Pinatubo did not erupt again after the major eruption on June 15, yet the three algorithms show retrieved SO$_2$ mass increases on June 17 and June 20 (the PCA and KK retrievals also indicate a small increase on June 18). Guo et al. (2004) attribute these increases to the sequestering of volcanic SO$_2$ by ice-ash mixtures in the cloud. They propose the sequestered SO$_2$
was released at a later time through sublimation of ice in the lower stratosphere. The oxidation of hydrogen sulfide offers another mechanism to account for the observed SO$_2$ mass increases in the first few days following the eruption. The combined results of the three algorithms support the conclusions of Guo et al., (2006) that the observed SO$_2$ mass increases in the temporal evolution of the plume are real.

Overall, the PCA retrieved ~3% more total SO$_2$ mass than MS_SO2. These differences are attributed to differences in how the algorithms handle aerosols and the differences in the calculated area of the SO$_2$ cloud due to the differences in the SO$_2$ values near the sensitivity threshold (~15 DU). Ash, sulfates and high SO$_2$ amounts impact the ozone retrieval, for as was seen in 3.2, systematic errors in SO$_2$ are anticorrelated with errors in O$_3$ (see Fig. S1). For the case of the KK algorithm, the total ozone


retrieved inside the SO₂ plume can be unrealistically low, and even negative in an extreme event like Mt. Pinatubo shown in Figures S3 and S4 of the supplement. The Figure S3 compares the KK ozone retrieval with MS_SO2 step 2 ozone retrieval and the Figure S4 compares scatterplots of SO₂ and total ozone for June 17 and June 18.

Figure 10. Daily SO₂ maps retrieved during Nimbus7 TOMS overpasses of the Pinatubo eruption cloud between June 15 and June 21, 1991, show excellent agreement between the two independent MS_SO2 and PCA retrieval algorithms.



Table 4 compared estimates of the $SO_2$ cloud area for the MS_SO2 and PCA retrievals. The area is most sensitive to the minimum detection threshold around the edges of the $SO_2$ cloud. The MS_SO2 and the PCA algorithms were directly compared by computing the sum of all the pixels where $\Sigma > 15$ DU (Fig. 10). For the six-day study period, the plume increased in size from about a little over 2 x $10^6$ km$^2$ to ~9 x $10^6$ km$^2$. The PCA retrieval observed a larger cloud area for the five of the six days, but the observed differences were within 10%. On June 16, shortly after the major eruption of June 15, the PCA estimated area for the $SO_2$ cloud is about ~15% greater than that retrieved using the MS_SO2 algorithm. The fresh plumes are opaque, which result in underestimating of $SO_2$ mass by all BUV algorithms due to the mixing of water, ice and aerosols (Krotkov et al., 1997). The PCA retrieval appears more sensitive to $SO_2$ near the edges of the cloud, where the ash aerosol loading is high (AI > 1.5). It should be noted that the soft calibration applied to the 340 nm channel, described in 3.3, may also contribute to the lowering the sensitivity around the edges of the plume. This correction effectively lowered the background $SO_2$ by about 3 DU.

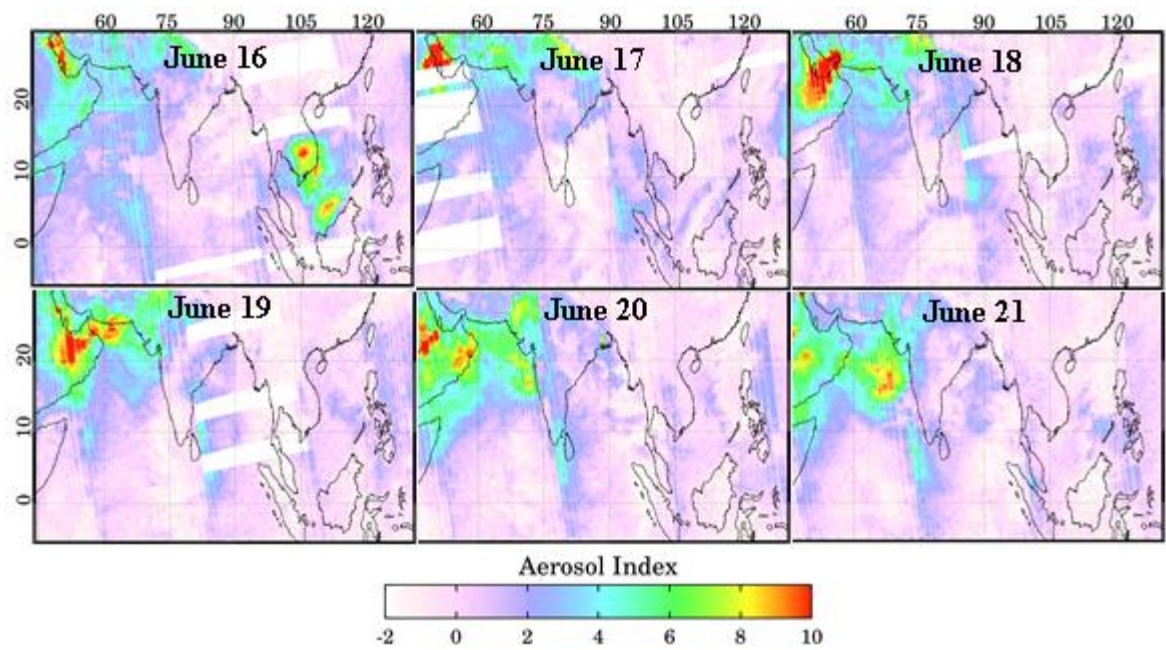

Figure 11. Daily Aerosol Index (AI) imagery retrieved using MS_SO2 between June 16 and June 21, 1991. Positive AI values over India and Arabia peninsula are due to dust aerosols, not related to the Pinatubo ash cloud.




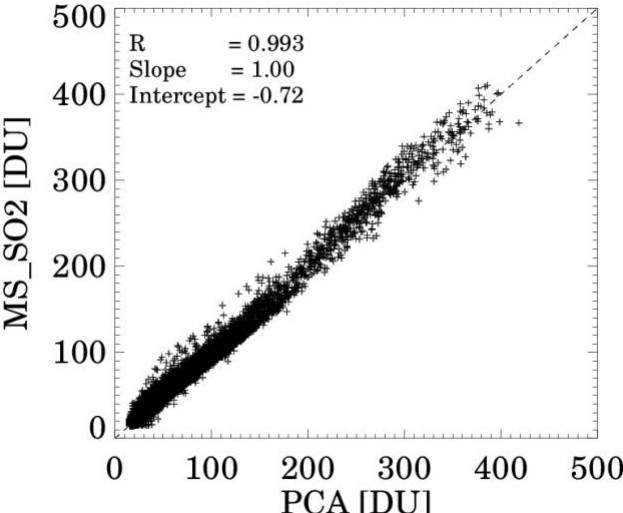

Figure 12. Scatterplot of retrieved SO₂ (Dobson Units) using PCA and MS_SO2 algorithms for the period June 15-21, 1991.

**Table 3**: Daily SO₂ mass and maximal SO₂ values for MS_SO2, PCA and KK (Guo et al., 2004) algorithms for the five days following the Pinatubo eruption on June 15, 1991.

| Day in June 1991 | MS_SO2 algorithm | | PCA algorithm | | Krueger_Kerr Algorithm | | Percent Difference (%) | |
|---|---|---|---|---|---|---|---|---|
| | SO₂ Mass (kt) | Max SO₂ (DU) | SO₂ Mass (kt) | Max SO₂ (DU) | SO₂ Mass (kt) | Max SO₂ (DU) | PCA | KK |
| 06/16 | 9.8 | 410 | 10.0 | 418 | 12.0* | 537 | -2.0 | 24.3 |
| 06/17 | 12.1 | 389 | 12.1 | 399 | 13.0 | 423 | 0.0 | 7.4 |
| 06/18 | 12.0 | 279 | 12.4 | 280 | 13.1* | 350 | 3.3 | 9.2 |
| 06/19 | 10.9 | 173 | 11.6 | 180 | 11.4* | 207 | 6.2 | 4.6 |
| 06/20 | 12.6 | 148 | 13.2 | 157 | 12.2 | 180 | 4.7 | -4.0 |
| 06/21 | 11.8 | 125 | 12.5 | 130 | 11.9 | 137 | 5.9 | 0.8 |

* Guo et al.,(2004) interpolated values in the missing data region seen in maps for June 16, 18, and 19



**Table 4**: SO₂ plume area and the number of FoVs where the retrieved SO₂ exceeded 15 DU for the MS_SO2, PCA and KK algorithms for the five days following for the Pinatubo eruption on June 15, 1991.

| | MS_SO2 | | PCA | | Percent Difference (%) |
|---|---|---|---|---|---|
| Day | Area ($\times 10^6$ km²) | NFovs ($\Sigma$>15 DU) | Area ($\times 10^6$ km²) | NFovs ($\Sigma$>15 DU) | PCA |
| 06/16 | 2.13 | 442 | 2.48 | 519 | 15.2 |
| 06/17 | 4.19 | 1006 | 4.04 | 971 | -3.6 |
| 06/18 | 5.05 | 1062 | 5.31 | 1088 | 5.0 |
| 06/19 | 5.09 | 910 | 5.30 | 957 | 4.0 |
| 06/20 | 7.27 | 1407 | 7.59 | 1487 | 4.3 |
| 06/21 | 8.44 | 1674 | 9.02 | 1805 | 6.6 |

## 6 Conclusions

This paper describes, a multi-satellite, multi-spectral UV algorithm (MS_SO2) for retrieving volcanic SO₂ that was used operationally to re-process measurements from the heritage Nimbus-7 TOMS and the current Deep Space Climate Observatory Earth Polychromatic Imaging Camera (Carn et al., 2018; Marshak et al., 2018). The MS_SO2 algorithm can process data from current hyperspectral UV spectrometers, such as SNPP/OMPS and Aura/OMI, using a convolved, discrete set of wavelengths, offering a viable means for intercomparing volcanic SO₂ retrievals from different missions.

We estimated random (noise) and systematic errors, related to the effects of volcanic aerosols and uncertainties in SO₂ height and partly corrected for absorbing ash, using positive aerosol index (AI) as a proxy for applying a Step 2 correction to the SO₂ retrievals. The correction could still underestimate SO₂ mass during the first days after extremely large eruptions (VEI > 3) due to BUV saturation. In such cases we recommend estimating e-folding time of the SO₂ conversion to sulfate aerosols, using later measurements and extrapolating SO₂ mass exponentially back in time to the eruption day (Krotkov et al., 2010).





The TOMS observational System Simulation Experiment (OSSE) confirmed unbiased SO₂ retrievals using synthetic radiances for $SO_2 < 100$-150 DU, but low biases for larger SO₂ amounts due to the presence of ash and sulfate aerosols. Therefore, operational MS_SO2 retrievals should provide a low boundary constraint on the SO₂ mass injected into the atmosphere from large eruptions. The

algorithm can be further improved by explicitly accounting for volcanic ash and sulfate aerosols, which was not feasible in the operational re-processing.

The MS_SO2 retrieval is also sensitive to the differences between the a priori and actual SO₂ center of mass altitude. Since this key parameter is not retrieved, the TOMS SO₂ product provides separate SO₂ column amounts assuming three different altitudes (8, 13 and 18 km). Users should select

one SO₂ column amount that is most appropriate for a particular eruption.

To validate the accuracy of the TOMS SO₂ retrievals, we compared the MS_SO2 with the independent measurements-bbased Principal Component Analysis (PCA) algorithm for the first six days following the 1991 Pinatubo eruption. The daily time series of the SO₂ retrievals show high correlation ($R^2 =0.986$) and unbiased agreement between the two retrievals over a broad SO₂ range between 15 DU

and 400 DU. We also compared the total SO₂ mass, peak SO₂ amounts and plume area with the heritage KK algorithm. This 3-way comparison showed the SO₂ mass within 10% for all days, except on June 16, when the Krueger-Kerr algorithm retrieved 24% higher SO₂ mass. This could be explained by interpolation over a region of missing TOMS measurements on June 16 (Guo et al., 2004). The remaining differences between the MS_SO2 and the PCA algorithms (3-7%) are attributed to the

differences in handling of aerosols, and different sensitivity thresholds of the algorithms.

The re-processed Nimbus-7 TOMS volcanic SO₂ data set, TOMSN7SO2 (Krotkov et al., 2017) is now publicly available through the Goddard Earth Sciences Data and Information Services Center (GES DISC) as part of the NASA's Making Earth System Data Records for Use in Research Environments (MEaSUREs) program at https://doi.org/10.5067/MEASURES/SO2/DATA202. We plan

to reprocess all follow-up multi-spectral UV (TOMS) and hyperspectral UV (OMI, OMPS) missions (Figure 1) with MS_SO2 and PCA algorithms to keep updating our multi-satellite volcanic SO₂ mass database (MSVOLSO2L4) archived at GES DISC (Carn 2019). It is important to continue quantifying SO₂ emissions from small explosive eruptions, as they may, collectively, play an important role in sustaining the persistent, background stratospheric aerosol layer, which is an important factor in global

climate forcing.




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
