# Peer review of "A new discrete wavelength BUV algorithm for consistent volcanic SO2 retrievals from multiple satellite missions"

_Atmospheric Measurement Techniques, 2019_

## Referee Comment (RC1) · Anonymous Referee #2 · 6 Jun 2019

General comments: In this paper the authors present a new algorithm for retrieving volcanic sulfur dioxide total columns from UV satellite instruments which is used operationally to process TOMS and EPIC data. It is also able to process data from current hyperspectral UV spectrometers. The algorithm has been applied to several volcanic cases and compared to a modified operational OMI & OMPS PCA algorithm.

The main advantage of such an algorithm is that it helps in assembling a long-term consistent satellite-based volcanic SO2 emissions climatology. Furthermore, this new algorithm is able to correctly retrieve SO2 even in the presence of aerosols using a 2-step procedure.

[Figure]

Overall I think the paper is suitable for publication in AMT after some moderate revisions. The paper can be slightly shortened in my opinion - although sections 2.1 and 2.2 are very interesting to read, they can be shortened and only focus on how they relate to the new MS_SO2 algorithm (i.e. remove the 'history' part of the algorithms).

What I am missing in the paper is a clear statement about the advantage of the new algorithm over e.g. the modified PCA algorithm the authors are using for comparison. Furthermore, a better description of how exactly the algorithm is working is required from my point of view (see below)

Specific comments: -P3 L15 Suggest to add S5P to the list

-P5 L3: BUV appears for the first time, please add the full name here

-P5 L3/4: Please add the wavelength of the three TOMS channels (i.e. move them here from line 7) Equation 2 and P8, L1-4: Usually the AMF corrects for the geometric optical path (as well as surface properties), as described, so why do the coefficients a and b depend on the satellite viewing geometry and cloud-surface properties as well?

-P8 L17: What are 'standard' O3 profiles? Please add a reference which profiles are used (e.g. TOMS V7...)?

-P8 L30+: I don't really understand what you are doing exactly. Are I0,I1,I2 the radiances at the three wavelengths? What is T? Temperature? Temperature for what? What is S_b? This appears here for the first time. Please add more details

-Figure 4 c/d Choose a different color bar (or color bar max values) since the SO2 VCD extends up to 550DU)

-Figure 5: After the correction, still a bias of about 1DU is visible in b). Why? Please add a plot showing the total SO2 map after the correction. Please also choose a different colorbar, with a white color in the center, such that is easier to identify positive and negative total columns (or differences)
-Section 4.1 I guess that the random errors change over time (degradation of the instrument), so it would be better to show and analyze the standard deviation as a function of time and not for the entire 10yrs time frame

-P22 L10 & Equation 11: So far you used the SUM symbol for the SO2 total column and OMEGA for O3. Please stick to that and don't use OMEGA_SO2 here. This is confusing

-P22 L21: Please describe your criterion why you are using only 5 PCs and not more (or less)

- Supplement P2 L31-32: What are the parameters I0, T and s_b? Please explain (see also my comment above)

Technical corrections: -P2, L27: SO2 (wrong format)

-Figure 1: European Sentinel-5P -> ESA Sentinel-5P

-P12,L26: (step 1) (missing parenthesis)

- Supplement Figure S3: Please remove the border around the colorbar

---

## Referee Comment (RC2) · Anonymous Referee #1 · 10 Jun 2019

Review of Fisher, et al., A new discrete wavelength BUV algorithm for consistent volcanic SO2 retrievals from multiple satellite missions.

I encourage publication of this paper because it represents a step forward in characterizing a factor in global climate change, is generally well-written, has a comprehensive review of background work, references the important publications, and describes all the procedures used to characterize the retrieval results, including simulations for error analysis. Exceptions needing correction are stated below.

This paper is significant because for the first time the entire four-satellite TOMS record since 1978 has been processed for SO2 column amounts and made available for analysis. The new algorithm can further extend the data record using the next generations of UV mapping instruments following TOMS. The existing TOMS volcanic record was constructed by processing individual eruptions that had been reported or accidentally detected. The analysis involved manual selection of a limited geographical region containing the plume because of long data processing times. This new dataset will allow a nearly complete census of essentially all eruptions of climate significance.

Earth satellites offered the first platforms for observation and measurement of the largest explosive volcanic eruption plumes. The ash clouds could usually be identified in AVHRR visible light images. However, the total erupted mass could best be obtained by measuring the quantity of absorbing gas in the eruption cloud. Sulfur dioxide was a volcanic gas that was rare in the atmosphere. Anomalous ozone retrievals from the Nimbus-7 TOMS instrument over Mexico were diagnosed as sulfur dioxide interference in the 1991 eruption of El Chichon that absorbed the UV wavelengths used for ozone measurements. The six instrument wavelengths had been selected for global ozone retrievals without consideration of sulfur dioxide interference since that gas is not a permanent atmospheric component. Thus, the task of discriminating sulfur dioxide from ozone absorption was not easy.

This paper clearly documents the evolution of TOMS ozone algorithms and the ad hoc ones developed to discriminate $SO_2$ from ozone absorption in nadir observations. This history is useful because after 30 years the background of current work tends to get lost.

The original Krueger $SO_2$ algorithm assumed that total ozone was unperturbed by the volcanic cloud and could be interpolated from extra-plume regions in TOMS traces across the cloud. Then sulfur dioxide was computed from the residual radiance at absorbed wavelengths.

In the succeeding algorithm, four parameters - sulfur dioxide, ozone, aerosol index (a measure of non-Raleigh scattering and aerosol absorption), and surface reflectivity -
were retrieved by inverting a 4 x 4 matrix. It was adapted for satellites from the algorithm that Jim Kerr had produced for Brewer Spectrophotometer data on overhead drifting volcanic clouds. This Krueger - Kerr method produced reasonable SO2, AI, and surface reflectivity results and was used routinely for many years to measure the position and sizes of eruption SO2 and ash clouds. The simultaneous observations of the sulfur dioxide and ash clouds showed that the ash would separate and fall out in a few days while the sulfur dioxide cloud drifted on, even in the largest eruption clouds. However, the model was too simple for accurate ozone retrievals and the iterative table search procedure used too much computer time so that it was executed only for select geographic regions that included the volcanic plume. Thus the entire TOMS dataset was never processed for sulfur dioxide until the work described in this paper was conducted.

The new MS_SO2 algorithm takes advantage of the operational TOMS ozone retrieval, TOMRAD radiative transfer look-up-tables that now include Jacobians, and the methodology of the K - K algorithm. The fixed absorption coefficients in the K-K matrix have been replaced by Jacobians, presumably resulting in faster processing times. Only the longest four of the six TOMS wavelengths are used based on a review of deviations of linearity in large SO2 and ash loadings at the shorter wavelengths.

The most error-prone SO2 retrievals are from very fresh major eruption plumes filled with water aerosols, sulfate, ash, sulfur dioxide, and other ejecta. The TOMS observations have shown that these situations persist only from hours to a couple of days as the ash falls out from the gas cloud and drifts away with the underlying wind field. After that the retrieval is much simpler given a nearly Raleigh-scattering atmospheric column. The MS_SO2 algorithm adds a Step 2 retrieval that is used when when the early complex plume conditions are detected by co-located large SO2 anomalies and high AI values. In this case the ozone is specified by interpolation across the cloud rather than computed, like the original SO2 algorithm.

Sect. 4.2 Systematic errors. . .

This paper estimates the retrieval errors due to lack of information about the height of the plume through modeling of the dependence of the Jacobians on plume height. The UV radiances contain no direct information about the height of the plume, so the uncertainties due to a wrong height assumption are important. They appear to be reasonably small except for high latitude plumes. However, a reference to non-existent Tables 1,2, and 3 needs to be corrected.

Sect 4.2.2. The aerosol sensitivity analysis using an OSSE is a useful tool for characterizing the range of linearity for the retrieval. Curiously both sulfate and absorbing aerosols produce the same negative deviations suggesting that non-Rayleigh scattering is more important than absorption.

Sect. 5. Comparison with PCS SO2 retrievals.

"Validation" of the MS_SO2 data with results from a new TOMS-adapted PCA algorithm seems to be a stretch. A "comparison" of results is certainly appropriate. However, this is not a validation in the usual sense of the word because the TOMS PCA algorithm is not itself validated. The hyperspectral PCA algorithm is well documented, and it should be straightforward to verify that the TOMS-PCA version produces the same result for an eruption covered by both multi-spectral and hyperspectral instruments. If there is documentation on this new 5 PC algorithm and data then it should be referenced. Or, if the changes can be shown to be inconsequential that should be stated. The authors should consider documenting the new PC algorithm is it does not already exist.

Having said that, the agreement between data from the two discrete band algorithms is impressive. This brings up new questions: What are the reasons to prefer one algorithm over the other? Is the MS_SO2 algorithm the best for processing the dataset?

Sect 5.1 Pinatubo eruption. Figure 11 shows AI maps that are not very useful due to confusion with unrelated dust clouds. Which are the ash clouds? It would be more useful to overlay SO2 images with AI images as that would illustrate the separation of the two eruption components.

Specific comments:

p11, line 20. ….in which large O3 anomalies…. do you mean large SO2 anomalies? On the next page you refer to either SO2 > 15 DU … or AI > 6. Please check for consistency.

p 12, line 6. "Step 2" instead of step 2. Note the other cases in the same section where "Step" is not capitalized.

p 18, line28. Tables 1, 2, and 3 do not exist.

p 21, line11. delete "an" in "The no aerosol case confirms an unbiased SO2 retrievals...".

p 22, line22. Define SLER.

Appendix

P 2, line 10. Eqs. (15a,b) could not be found.

---

## Author Comment (AC1) · 7 Aug 2019

Anonymous Referee #1 Review of Fisher, et al., A new discrete wavelength BUV algorithm for consistent volcanic SO2 retrievals from multiple satellite missions. We would like to thank the reviewer for nicely characterizing our work, its relevance, and for helping to improve the readability of the paper. I encourage publication of this paper because it represents a step forward in characterizing a factor in global climate change, is generally well-written, has a comprehensive review of background work, references the important publications, and describes all the procedures used to characterize the retrieval results, including simulations for error analysis.

[Figure]

Exceptions needing correction are stated below. This paper is significant because for the first time the entire four-satellite TOMS record since 1978 has been processed for SO2 column amounts and made available for analysis. The new algorithm can further extend the data record using the next generations of UV mapping instruments following TOMS. The existing TOMS volcanic record was constructed by processing individual eruptions that had been reported or accidentally detected. The analysis involved manual selection of a limited geographical region containing the plume because of long data processing times. This new dataset will allow a nearly complete census of essentially all eruptions of climate significance. Earth satellites offered the first platforms for observation and measurement of the largest explosive volcanic eruption plumes. The ash clouds could usually be identified in AVHRR visible light images. However, the total erupted mass could best be obtained by measuring the quantity of absorbing gas in the eruption cloud. Sulfur dioxide was a volcanic gas that was rare in the atmosphere. Anomalous ozone retrievals from the Nimbus-7 TOMS instrument over Mexico were diagnosed as sulfur dioxide interference in the 1991 eruption of El Chichon that absorbed the UV wavelengths used for ozone measurements. The six instrument wavelengths had been selected for global ozone retrievals without consideration of sulfur dioxide interference since that gas is not a permanent atmospheric component. Thus, the task of discriminating sulfur dioxide from ozone absorption was not easy. This paper clearly documents the evolution of TOMS ozone algorithms and the ad hoc ones developed to discriminate SO2 from ozone absorption in nadir observations. This history is useful because after 30 years the background of current work tends to get lost. The original Krueger SO2 algorithm assumed that total ozone was unperturbed by the volcanic cloud and could be interpolated from extra-plume regions in TOMS traces across the cloud. Then sulfur dioxide was computed from the residual radiance at absorbed wavelengths. In the succeeding algorithm, four parameters - sulfur dioxide, ozone, aerosol index (a measure of non-Raleigh scattering and aerosol absorption), and surface reflectivity - were retrieved by inverting a 4 x 4 matrix. It was adapted for satellites from the algorithm that Jim Kerr had produced for Brewer Spectrophotometer data on overhead drifting volcanic clouds. This Krueger - Kerr method produced reasonable SO2, AI, and surface reflectivity results and was used routinely for many years to measure the position and sizes of eruption SO2 and ash clouds. The simultaneous observations of the sulfur dioxide and ash clouds showed that the ash would separate and fall out in a few days while the sulfur dioxide cloud drifted on, even in the largest eruption clouds. However, the model was too simple for accurate ozone retrievals and the iterative table search procedure used too much computer time so that it was executed only for select geographic regions that included the volcanic plume. Thus the entire TOMS dataset was never processed for sulfur dioxide until the work described in this paper was conducted. The new MS_SO2 algorithm takes advantage of the operational TOMS ozone retrieval, TOMRAD radiative transfer look-up-tables that now include Jacobians, and the method- ology of the K - K algorithm. The fixed absorption coefficients in the K-K matrix have been replaced by Jacobians, presumably resulting in faster processing times. Only the longest four of the six TOMS wavelengths are used based on a review of deviations of linearity in large SO2 and ash loadings at the shorter wavelengths. MS_SO2 does not use the 312 (shortest) or the 360 (second longest) nm channels. The most error-prone SO2 retrievals are from very fresh major eruption plumes filled with water aerosols, sulfate, ash, sulfur dioxide, and other ejecta. The TOMS observations have shown that these situations persist only from hours to a couple of days as the ash falls out from the gas cloud and drifts away with the underlying wind field. After that the retrieval is much simpler given a nearly Raleigh-scattering atmospheric column. The MS_SO2 algorithm adds a Step 2 retrieval that is used when the early complex plume conditions are detected by co-located large SO2 anomalies and high AI values. In this case the ozone is specified by interpolation across the cloud rather than computed, like the original SO2 algorithm. Sect. 4.2 Systematic errors. . . This paper estimates the retrieval errors due to lack of information about the height of the plume through modeling of the dependence of the Jacobians on plume height. The UV radiances contain no direct information about the height of the plume, so the uncertainties due to a wrong height assumption are

important. They appear to be reasonably small except for high latitude plumes. However, a reference to non-existent Tables 1,2, and 3 needs to be corrected. This error in the numbering was corrected. These tables (S1, S2 and S3) are in the supplement. Sect 4.2.2. The aerosol sensitivity analysis using an OSSE is a useful tool for characterizing the range of linearity for the retrieval. Curiously both sulfate and absorbing aerosols produce the same negative deviations suggesting that non-Rayleigh scattering is more important than absorption. We agree that this is an interesting result. We don't see a large bias in SO2 retrievals from aerosols for SO2 < 150 DU. It should also be noted that between 100 and 150 DU, there is a slight positive bias in Fig. 9 (middle panel: "sulfates only"). For SO2 > 150 DU, the bias is clearly negative. Sect. 5. Comparison with PCS SO2 retrievals. "Validation" of the MS_SO2 data with results from a new TOMS-adapted PCA algorithm seems to be a stretch. A "comparison" of results is certainly appropriate. However, this is not a validation in the usual sense of the word because the TOMS PCA algorithm is not itself validated. The hyperspectral PCA algorithm is well documented, and it should be straightforward to verify that the TOMS-PCA version produces the same result for an eruption covered by both multi-spectral and hyperspectral instruments. If there is documentation on this new 5 PC algorithm and data then it should be referenced. Or, if the changes can be shown to be inconsequential that should be stated. The authors should consider documenting the new PC algorithm is it does not already exist. Thank you for pointing this out. We agree that the word "comparison" is more appropriate. We have changed the wording in the revised abstract and section 5 of the manuscript. The 5 wavelength PC TOMS algorithm is an experimental algorithm that has only been tested with Nimbus-7 TOMS. As Nimbus-7 data record ended before hyperspectral data became available, for this study we are not able to directly compare PCA SO2 retrievals between multi-spectral and hyperspectral instruments (or implementations). In the future, we will document the experimental 5 PC algorithm and implement it with different TOMS instruments. Having said that, the agreement between data from the two discrete band algorithms is impressive. This brings up new questions: What are the reasons to prefer one algorithm over the other? Is the MS_SO2 algorithm the best for processing the dataset? As mentioned above, the PCA algorithm is experimental and has only been tested with Nimubs-7 for a few selected eruptions. One advantage of the MS_SO2 algorithm is that it can produce both O3 and SO2 retrievals, whereas PCA algorithm requires O3 from another algorithm as an input. The MS_SO2 algorithm also retrieves spectral dependence of LER, dR/dïĄň, which can be used to compute an aerosol index. The additional parameters provide more information for better assessing the quality of the retrieval of column SO2. The retrieval of ozone and the aerosol index (computed using retrieval of dR/ dïĄň) are especially useful in the analysis of the results from MS_SO2. Also, since MS_SO2 can be easily configured for any BUV satellite, it can be used as a transfer standard in assessing the differences between SO2 retrievals from different satellites, including hyperspectral instruments. The PCA, on the other hand, was designed to exploit the spectral information contained in hyperspectral measurement by using a large number of PCs to separate the variance introduced into the retrieval by a multitude of physical and instrumental parameters. When applied to measurements from hyperspectral instruments, the PCA has sensitivity down to 0.5 DU.

Sect 5.1 Pinatubo eruption. Figure 11 shows AI maps that are not very useful due to confusion with unrelated dust clouds. Which are the ash clouds? It would be more useful to overlay SO2 images with AI images as that would illustrate the separation of the two eruption components. We followed the reviewer's suggestion and produced a new figure showing AI maps with a contour of the SO2 plume superimposed. We also reduced the geographical boundaries of the map to better show the evolution of the plume. We used the same geographical boundaries in the two figures showing the SO2 and the AI maps from June 16 to June 21. The geographical range remains fixed for each day.

Replies to Specific Comments by Reviewer 1 Specific comments: -P3 L15 Suggest adding S5P to the list We added EU/ESA Copernicus S5P to the list on P3. -P5 L3: BUV appears for the first time, please add the full name here Spelled out name of

acronym: Backscattered Ultraviolet on P5 -P5 L3/4: Please add the wavelength of the three TOMS channels (i.e. move them here from line 7): Added wavelengths of the three absorbing TOMS Channels. Moved them to line 7. - Equation 2 and P8, L1-4: Usually the AMF corrects for the geometric optical path (as well as surface properties), as described, so why do the coefficients a and b depend on the satellite viewing geometry and cloud-surface properties as well? The AMF = SCD/VCD converts "slant column density (SCD)" into the vertical column density (VCD) of absorbing gases and depends on the viewing geometry, surface albedo, clouds and aerosols. In equation (2) we used the geometric AMF approximation: AMF~Sg=sec(SZA) + sec(VZA) (Krueger et al., 1995). However, the polynomial coefficients a and b in a scattering atmosphere also vary with observational geometry, surface pressure and reflectivity, clouds and aerosol properties (Krotkov et al., 1997). Indeed, we later developed a method for retrieving volcanic ash optical depth and single scattering albedo by inverting measured a and b parameters (Fig. 9 in Krotkov et al.,(1997).

-P8 L17: What are 'standard' O3 profiles? Please add a reference which profiles are used (e.g. TOMS V7. . .)? TOMRAD RTM requires ozone (primary gas) and secondary gas (SO2) profiles in calculating the BUV radiances at the TOA. The standard O3 profiles are a priori ozone profiles that vary with total ozone and latitude. These profiles provide essential input for the forward model. We have inserted two references (Klenk et al., 1983 and OMI Algorithm Theoretical Basis Document, vol. II, 1997) to further support the motivation for using them.

-P8 L30+: I don't really understand what you are doing exactly. Are I0, I1, I2 the radiances at the three wavelengths? What is T? Temperature? Temperature for what? What is Sb? This appears here for the first time. Please add more details. We moved Eq. 3 to the supplement and added explanation for each parameter (T, Sb). Basically, the equation 3 allows analytical calculation of BUV radiance using pre-computed radiative transfer parameters (Sb, T, I0), stored in the LUTs used in the operational algorithm. The equation and the terms are now more fully explained in the supplement

(T = Transmittivity, Sb = spherical reflectivity of the atmosphere).

-Figure 4 c/d Choose a different color bar (or color bar max values) since the SO2 VCD extends up to 550DU) Extended range of color bar on imagery displayed in Figures 4 c & d. -Figure 5: After the correction, still a bias of about 1 DU is visible in b). Why? We don't necessarily expect a zero global bias, for 1) the correction is computed as a function of cross track position and 2) the spatial domain on which the correction is based is constrained to clean regions of the Pacific Ocean. Moreover, the N-value correction also displays a small latitudinal dependence with respect to the tropics and mid-latitudes, however, we found that the differences in these two latitudinally broad regions are less than the noise of the retrieval and so we assume no latitudinal dependence in applying the correction.

- Please add a plot showing the total SO2 map after the correction. Please also choose a different color bar, with a white color in the center, such that is easier to identify positive and negative total columns (or differences) We removed the global SO2 noise maps (figs b, c, d) that showed the effects of applying the soft calibration correction to all the data. The effect of a 3 DU shift on the SO2 background is discernable but very small, so we decided to only present the first plot showing the change in the pdf before and after. We further combined the two pdf plots into a single plot and revised the paragraph describing new figure 5. We also moved Fig. 6 to the supplement in association with the description of our soft calibration methodology using the 340 nm channel.

-Section 4.1 I guess that the random errors change over time (degradation of the instrument), so it would be better to show and analyze the standard deviation as a function of time and not for the entire 10yrs time frame We agree that degradation of the instrument and time-dependent calibration drift can be characterized in the way suggested. We will consider this characterization in our future work. In this section, we intended to compare the random errors and consequently, the sensitivity of Nimbus 7/TOMS to a more modern hyperspectral instrument. To examine the time dependence of the

random errors over the mission, we would need to include an additional section and figure to the paper. We might be able to more pointedly mention the long-term calibration of the instrument (which has been documented for ozone retrievals) and provide a reference to some of ozone papers/TOMS documentation that have probed this issue.

-P22 L10 & Equation 11: So far you used the SUM symbol for the SO2 total column and OMEGA for O3. Please stick to that and don't use OMEGA_SO2 here. This is confusing. Agreed. Fixed these symbols. -P22 L21: Please describe your criterion why you are using only 5 PCs and not more (or less) As TOMS only has 6 wavelengths, the maximum number of PCs can be obtained from the PCA analysis on the radiance data is 6. For inversion of radiances (from all six channels) for SO2 amounts, the maximum number of PCs can be used is 5, as the SO2 Jacobians spectrum also needs to be included in the inversion. We also tested retrievals using just 3 or 4 PCs and found much larger artifacts, so we elect to use 5 PCs in the TOMS PCA SO2 retrieval algorithm.

- Supplement P2 L31-32: What are the parameters I0, T and sb? Please explain (see also my comment above) Moved eq. 3 and text describing this equation from paper to supplement. Added complete definitions and explanation.

Technical corrections: -P2, L27: SO2 (wrong format): Fixed. -Figure 1: European Sentinel-5P -> ESA Sentinel-5P Fixed. -P12, L26: (step 1) (missing parenthesis) - Supplement Figure S3: Please remove the border around the color bar Removed border around color bar on S3.

Please also note the supplement to this comment:
https://www.atmos-meas-tech-discuss.net/amt-2019-150/amt-2019-150-AC1-supplement.pdf

**Supplement:**

General comments: In this paper the authors present a new algorithm for retrieving volcanic sulfur dioxide total columns from UV satellite instruments which is used operationally to process TOMS and EPIC data. It is also able to process data from current hyperspectral UV spectrometers. The algorithm has been applied to several volcanic cases and compared to a modified operational OMI & OMPS PCA algorithm.

The main advantage of such an algorithm is that it helps in assembling a long-term consistent satellite-based volcanic SO2 emissions climatology. Furthermore, this new algorithm is able to correctly retrieve SO2 even in the presence of aerosols using a 2-step procedure.

Overall, I think the paper is suitable for publication in AMT after some moderate revisions. The paper can be slightly shortened in my opinion - although sections 2.1 and 2.2 are very interesting to read, they can be shortened and only focus on how they relate to the new MS_SO2 algorithm (i.e. remove the 'history' part of the algorithms).

> *We decided to leave these sections in the paper. We feel that the historical connections of MS_SO2 to the development of the TOMS ozone algorithm and to the early development of the Krueger-Kerr algorithm are important to the heritage of our algorithm.*

What I am missing in the paper is a clear statement about the advantage of the new algorithm over e.g. the modified PCA algorithm the authors are using for comparison. Furthermore, a better description of how exactly the algorithm is working is required from my point of view (see below)

> *The MS_SO2 algorithm retrieves 4 parameters (SO2, O3, $dR/d\lambda$ and R) while the PCA only retrieves $SO_2$. The additional parameters provide more physical information about the eruption that can be used to better assess the quality of the retrieval. The retrieval of ozone and the aerosol index (using retrieval $dR/d\lambda$) are especially useful in analyzing results from MS_SO2. We will clarify this point in the paper's conclusion.*

Specific comments:

p11, line 20. . . in which large $O_3$ anomalies. . .. do you mean large $SO_2$ anomalies?

> *No. We meant large $O_3$ anomalies. Identifying $O_3$ anomalies inside of the $SO_2$ plume is a key step in the step 2 algorithm. The O3 and associate SO2 anomalies are anti-correlated (Fig. S1). The correction to the $SO_2$ comes about via a correction to the $O_3$ for a given FoV. For the O3 anomaly to be clearly distinguished (and subsequently corrected), the $O_3$ anomalies inside the plume have to be well above the regional $O_3$ variability outside of the plume (along with other corroborating criteria ($SO_2$ and AI)).*

On the next page you refer to either $SO_2 > 15$ DU ... or AI > 6. Please check for consistency.

*Our selection criterion is designed to identify the $SO_2$ plume region (generally where $SO_2$ > 15 DU). Heavy ash loading of the column, however, produces a shielding effect that can result in an underestimate of the $SO_2$, causing some FoVs associated with the plume to be less than 15 DU (e.g., volcanic $SO_2$ around the boundaries of the plume where $SO_2$ ~15 DU).  So, in addition to testing all FoVs where $SO_2$ > 15 DU, we also consider a second independent condition that selects FoVs with super-high ash concentrations (AI > 6).  This loosens the strict requirement on $SO_2$ and allows for values less than 15 DU to be further tested. We first select FoVs meeting these two criteria and then decide whether or not to apply step 2 based on the size of the ozone anomaly for that FoV.*

p 12, line 6. "Step 2" instead of step 2. Note the other cases in the same section where "Step" is

not capitalized.

*Fixed. We decided to lower-case step1 and step2.*

p 18, line28. Tables 1, 2, and 3 do not exist.

*These 3 tables were moved to the supplement. Should have been labeled S1, S2 and S3.*

p 21, line11. delete "an" in "The no aerosol case confirms an unbiased SO2 retrievals...".

*Fixed.*

p 22, line22. Define SLER.

*Defined Simple LER, assuming clouds are at the surface (no Mixed LER, MLER, that separates cloudy and clear regions of the FoV based on cloud fraction). The SLER was described in 2.1 but the acronym was not used and so we define the acronym in 2.1.*

Appendix
P 2, line 10. Eqs. (15a, b) could not be found.
*Fixed numbering of equations in the supplement.*